# EXGRA-MED: Extended Context Graph Alignment for Medical Vision-Language Models

**Duy M. H. Nguyen** [1,2,3]    **Nghiem T. Diep** [1]*    **Trung Q. Nguyen** [1]*    **Hoang-Bao Le** [1]
**Tai Nguyen** [1]    **Tien Nguyen** [4,5]    **TrungTin Nguyen** [6,7]    **Nhat Ho** [9]    **Pengtao Xie** [10, 11]
**Roger Wattenhofer** [12]    **James Zou** [13]    **Daniel Sonntag** [1,8]†    **Mathias Niepert** [2,3]†

[1] German Research Centre for Artificial Intelligence (DFKI),
[2] Max Planck Research School for Intelligent Systems (IMPRS-IS), [3] University of Stuttgart,
[4] University Medical Center Gottingen, [5] Max Planck Institute for Multidisciplinary Sciences,
[6] ARC Centre of Excellence for the Mathematical Analysis of Cellular Systems,
[7] School of Mathematical Sciences, Queensland University of Technology,
[8] University of Oldenburg, [9] University of Texas at Austin,
[10] University of California San Diego, [11] MBZUAI, [12] ETH Zurich, [13] Stanford University

\* Co-second contribution    † Co-senior authors.
 Exgra-Med

## Abstract

State-of-the-art medical multi-modal LLMs (med-MLLMs), such as LLAVA-MED and BIOMEDGPT, primarily depend on scaling model size and data volume, with training driven largely by autoregressive objectives. However, we reveal that this approach can lead to weak vision-language alignment, making these models overly dependent on costly instruction-following data. To address this, we introduce EXGRA-MED, a novel multi-graph alignment framework that jointly aligns images, instruction responses, and extended captions in the latent space, advancing semantic grounding and cross-modal coherence. To scale to large LLMs (e.g., LLaMa-7B), we develop an efficient end-to-end training scheme using black-box gradient estimation, enabling fast and scalable optimization. Empirically, EXGRA-MED matches LLAVA-MED's performance using just 10% of pre-training data, achieving a 20.13% gain on VQA-RAD and approaching full-data performance. It also outperforms strong baselines like BIOMEDGPT and RADFM on visual chatbot and zero-shot classification tasks, demonstrating its promise for efficient, high-quality vision-language integration in medical AI.

## 1 Introduction

Generic Multi-Modal Large Language Models (MLLMs) such as GPT-4V [7], LLaVa [57], and Next-GPT [97] unify text, image, and audio processing for tasks like captioning and visual reasoning. A key component in training MLLMs is instruction-following (IF) data [60], which involves complex, often multi-turn interactions grounded in image content [88]. In the medical domain, specialized IF datasets, including medical images, clinical notes, and diagnostic criteria, have been curated to adapt general-purpose MLLMs while leveraging their pre-learned knowledge and minimizing training costs [98]. For example, LLaVA-Med [50] samples 600K image-text pairs from PMC-15M [106], using GPT-4 to generate around 60K multi-modal IF examples. The training involves two pretraining steps: (i) aligning vision encoders and language decoders via projection layers, and (ii) jointly training the model (excluding the vision encoder) on medical IF data using an auto-regressive objective. The resulting model is then fine-tuned for downstream medical tasks.

Following the above approach, most later works have focused on scaling up the amount of medical IF data [98, 104, 30] or increasing the model size by incorporating larger vision encoders or language

decoders [96, 39] while relying on the same standard autoregressive learning scheme. Contrary to this, we question the effectiveness of autoregressive objective functions when learning medical-MLLM with IF data. Surprisingly, *our findings reveal that autoregressive learning is highly data-hungry during pre-training*, i.e., without sufficient medical IF samples, model performance plummets for downstream tasks, *even after fine-tuning*. To illustrate this, we pre-trained LLAVA-Med using only $10\%$ of the data and compared it to the version trained on $100\%$. Both models were fine-tuned on two medical visual question-answering tasks - VQA-RAD [48] and PathVQA [31] - and their average performance on open- and close-ended questions are compared. The results show a dramatic decline: from $72.64\%$ to $52.39\%$ on VQA-RAD (Figure 1) and from $64.06\%$ to $56.15\%$ on PathVQA (Table 1). This underscores the instability of medical-MLLM trained with autoregressive methods and highlights the problem that these methods require the curation of enough medical IF data to achieve satisfactory performance.

To address the limitations of autoregressive training under limited instruction-following data, we propose EXGRA-MED, a novel multi-graph alignment framework that strengthens cross-modal understanding in multi-modal large language models (MLLMs). At the core of our approach is the construction of three modality-specific graphs: one for visual features extracted by a vision encoder, and two for different textual variants of the instruction. These graphs represent semantic relationships within and across modalities, and we formulate a combinatorial multi-graph alignment problem to learn consistent triplet-level associations between the image, its instruction, and a semantically enriched variant. This alignment objective is jointly optimized with the autoregressive language modeling loss, enabling the model to enhance semantic depth, coherence, and instruction-following ability. To generate the enriched instruction variant, we use a frozen LLM (GPT-4 [7]) to produce a contextually extended version of each instruction that highlights key concept relationships without altering the original intent. The

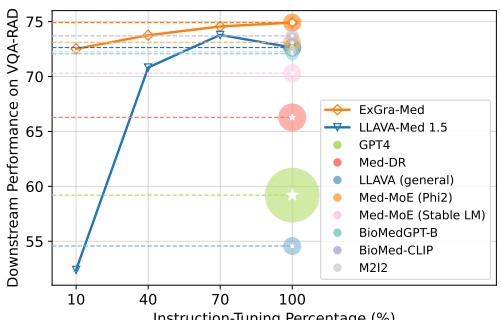

Figure 1: Our EXGRA-MED versus LLaVA-Med across varying instruction-following (IF) pre-training data sizes, **highlighting the data-hungry behavior of auto-regressive modeling**. Both *models are fine-tuned on the same VQA-RAD training set* after the pre-training stage at each IF rate. At 100% IF pre-training, ExGra-Med and LLaVA-Med are benchmarked against other state-of-the-art models, *all fine-tuned on the same VQA-RAD training set* (except GPT-4, which is evaluated without fine-tuning). Circle radius represents the number of model parameters.

vision encoder and language model (LLaMa [93]) then process the image, instruction, and extension independently to produce node embeddings used in the alignment step. Unlike naive data augmentation, our use of GPT-4 enriches supervision and facilitates fine-grained graph-based correspondence learning across modalities (Figure 3).

Our method stands apart from prior multi-modal alignment approaches for LLMs [72, 51, 16] in two key ways. (i) Instead of merely learning projection layers between frozen vision and language models, we *train the LLM directly* via a multi-graph alignment framework. (ii) We also *extend pairwise contrastive learning* by integrating global structural graph constraints, enabling alignment not just between individual image-caption pairs but across entire datasets. This graph-based design captures both feature and relational consistency, critical for handling similar entities in medical data. While multi-graph alignment is typically non-differentiable [80] and computationally intensive [74], we address these challenges using implicit maximum likelihood estimation [70, 63]. This enables efficient gradient-based training over large LLMs (e.g., LLaMa-7B) using a barycenter graph [8] for alignment, allowing our model to scale effectively while preserving strong alignment performance.

In summary, we make the following key contributions:

- We reveal the data-demanding nature of autoregressive modeling in pre-training medical-MLLM (LLaVa-Med), showing that insufficient instruction-following data leads to significant performance drops on downstream tasks, even after fine-tuning.

- We introduce a new multi-graph alignment objective that establishes triplet correlations among images, their instruction-following context, and their enriched versions. Additionally, we developed an efficient solver for training with LLMs and outlined theoretical properties related to distance and the shortest path in the geodesic space of multi-modal graphs.

- We empirically demonstrate that using a small amount of pre-training data, EXGRA-MED can achieve performance comparable to LLaVa-Med trained on $100\%$ data. Additionally, when trained on larger datasets, EXGRA-MED *outperforms* several state-of-the-art *med-MLLMs* and *multi-modal pre-training* algorithms across three Medical VQA tasks, medical visual chat, and the average zero-shot image classification performance on 23 datasets.

## 2   Related Work

**Medical Multi-Modal LLMs.** Recent developments in medical-MLLM like Biomed-GPT [104], MedFlamingo [64], Med-Dr [30], LLAVA-Med [50], and Med-PaLMs [83, 94] are transforming healthcare by integrating diverse data types and scaling medical instruction data. Biomed-GPT excels with multiple biomedical modalities, MedFlamingo focuses on few-shot learning for medical visual question answering, and LLAVA-Med leverages large-scale biomedical image-text pairs for improved performance. Commonly, these models emphasize scaling medical instruction data and increasing model parameters to enhance accuracy and applicability in real-world medical scenarios. In contrast, *our approach examines the widely used autoregressive pre-training algorithms* and demonstrates that incorporating enriched context multi-graph alignment of existing instruction samples can significantly enhance medical-MLLM performance without requiring larger models or extensive datasets.

**Visual Instruction Tuning.** Visual instruction tuning techniques aim to bridge the gap between frozen vision-language models and frozen LLMs trained on unimodal data, enabling them to work effectively in a multi-modal context. These methods involve (i) learning a multi-layer perceptron (MLP) layer to map embeddings from the vision model to the language model as LLaVa [57], VideoLLM [16]; (ii) using adapter-based adjustment as LLaMa-adapter [105], Voxposer [35], or (iii) learning multi-modal perceiver by gated cross-attention [9] or Q-Former as in BLIP-2 [51]. Pre-training algorithms to train these models can be combined with both auto-regressive and contrastive learning [72, 102] or image-text matching as in [52, 51]. Our algorithm differs from those by *focusing on directly training LLMs rather than lightweight projectors*. This requires a fast solver capable of efficiently handling forward and backward passes through large-scale LLMs with extensive parameters.

**Vision-language Pretraining Algorithm.** Pre-training algorithms commonly applied for vision-language models, like CLIP [77], employ various strategies. Among these, generative approaches are widely used, including masked prediction in language models [24, 85], or autoregressive algorithms that predict sequential text in LLMs [57, 105]. Another prominent direction focuses on discriminative methods, which learn contrastive distances between image-text pairs [58, 102, 45], optimal transport [15, 68], or impose clustering constraints [72]. *Toward multi-modal learning across three or more modalities*, there also exist works such as PAC-S[81], GeoCLAP [46], and IMAGEBIND [26], extending the CLIP or InfoNCE [71] to align embeddings across multiple modalities simultaneously.

Our function departs from these by *generalizing them into a combinatorial graph-matching formulation across cross-domain graphs*. While LVM-Med [62] is the most similar to our approach, it targets alignment within vision tasks, whereas we align images, instruction-following data, and extended contextual information. We provide a more comprehensive comparison between EXGRA-MED and related works in the Appendix.

**Scalable Multi-Graph Alignment.** Graph alignment across $K$ domains ($K \geq 3$) is highly computationally intensive. Current methods, such as multi-marginal optimal transport [54, 75], Wasserstein barycenters [69], and multi-adjacency matrix assumptions [13, 89], relax the problem but are limited to small-scale tasks and require multiple solver steps, making them inefficient for LLM training. In contrast, our algorithm leverages heuristic solvers [90, 80] and modern gradient estimation techniques for black-box optimization [70, 63], enabling scalable and efficient performance for large language models. A deeper analysis of this factor is discussed in the Ablation study.

## 3   Multi-graph Alignment Learning

We denote the vision encoder, projector, and large-language model (LLM) models are $f_\theta(.)$, $h_\phi(.)$, $g_\sigma(.)$, respectively. Figure 3 illustrates our EXGRA-MED algorithm, which learns parameters for these models by solving a triplet alignment between modalities in instruction tuning data. Below, we summarize the notations used before describing each component in detail.

**Notation.** Given any tensor $\mathbf{T} = (T_{i,j,k,l})$ and matrix $\boldsymbol{M} = (M_{k,l})$, we use $\mathbf{T} \otimes \boldsymbol{M}$ to denote the tensor-matrix multiplication, *i.e.,* the matrix $(\sum_{k,l} T_{i,j,k,l} M_{k,l})_{i,j}$. Given $\boldsymbol{Y} = [\boldsymbol{y}_1, \boldsymbol{y}_2, ..., \boldsymbol{y}_N] \in$

$\mathbb{R}^{N \times d}$, we define $\mathbb{E}(\boldsymbol{Y}) = \frac{1}{N} \sum_{i=1}^{N} \boldsymbol{y}_i \in \mathbb{R}^d$. Moreover, we define the matrix scalar (or inner) product associated with the Frobenius norm between two matrices $\boldsymbol{M} = (M_{i,j})$ and $\boldsymbol{N} = (N_{i,j})$ as $\langle \cdot, \cdot \rangle$, *i.e.,* $\langle \boldsymbol{M}, \boldsymbol{N} \rangle = \sum_{i,j} M_{i,j} N_{i,j}$. We write $[M] = \{1, 2, \ldots, M\}$ for any natural number $M$.

## 3.1 Extended context enriched medical instruction following data

Recent research has demonstrated that incorporating longer context significantly enhances LLMs' ability to process complex inputs and improves instruction-following by retaining more relevant information [59, 11, 73]. Building on this insight, we expand medical instruction-following data by generating *contextually enriched paraphrased versions of existing samples*, offering a complementary perspective to the original dataset. There are two key motivations for incorporating both original and extended captions in our multi-graph alignment framework. (i) First, aligning with original captions preserves precise, domain-specific details, while extended captions enhance semantic richness, leading to more robust image embeddings. (ii) Second, this approach helps the LLM generate contextually rich yet semantically consistent responses, improving alignment across diverse linguistic forms (Table 6).

In particular, a typical instruction sample includes $\{\boldsymbol{X}_v, [\boldsymbol{X}_q^1, \boldsymbol{X}_a^1], ..., [\boldsymbol{X}_q^L, \boldsymbol{X}_a^L]\}$ where $\boldsymbol{X}_v$ is an input image, $\boldsymbol{X}_q^l$ a question, and $\boldsymbol{X}_a^l$ an answer at round $l$ in multi-round $L$ of a conversation. In the medical domain, most of the questions are generic, and the information answer usually covers the question, so we only focus on extending the answer $\boldsymbol{X}_a$. We leverage the GPT API with a `prompt` to form an extended context for each $\boldsymbol{X}_a^l$ by:

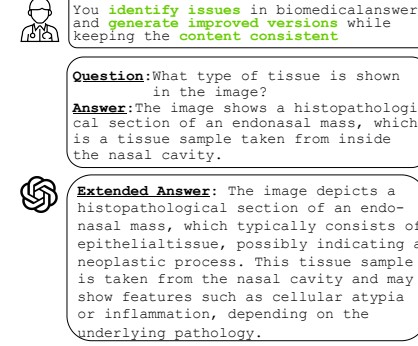

$$\boldsymbol{X}_{ae}^l = \text{GPT}\left(\boldsymbol{X}_q^l, \boldsymbol{X}_a^l, \text{prompt}\right), \ \forall l \in [L]. \quad (1)$$

The details for `prompt` are presented in the Appendix. In short, we ask GPT to provide additional explanations for concepts that appeared in the original answer $\boldsymbol{X}_a$ while keeping the content consistent. An example output for $\boldsymbol{X}_{ae}^l$ is illustrated in Figure 2 and Table 8. It's worth noting that other frozen LLMs like Gemini are also valid in our method (Table 4).

Figure 2: Illustration for creating the extended context instruction-following data powered by GPT-4o.

## 3.2 Multi-graph construction on vision-language embedding

Each *image* $\boldsymbol{X}_v \in \mathbb{R}^{3 \times H \times W}$, where $(H, W)$ are the original spatial dimensions, is divided it into a sequence of visual patches $\boldsymbol{U} = [u_i]_{i=1}^N$ with $N = (H \times W)/U$ with $U$ the patch size. Using a pre-trained ViT model $f_\theta$, we extract patch-wise features as $\boldsymbol{V} = f_\theta(\boldsymbol{U}) \in \mathbb{R}^{N \times d_v}$ and apply another projector to map it into the projected embedding $\boldsymbol{Z} = h_\phi(\boldsymbol{V}) \in \mathbb{R}^{N \times d}$. We then pool the features from the image patches to define a global description as $\boldsymbol{Z}_v = \mathbb{E}(\boldsymbol{Z}) \in \mathbb{R}^d$. For each *language input* $\boldsymbol{X}_c^l \in \{\boldsymbol{X}_a^l, \boldsymbol{X}_{ae}^l\}$ with $c \in \{a, ae\}$, we assume it has $M$ tokens, i.e., $\boldsymbol{X}_c^l = [\boldsymbol{x}_j]_{j=1}^M \in \mathbb{R}^M$, and feed it into the LLM model to extract a set of embedding $\boldsymbol{Z}_c^l = g_\sigma([\boldsymbol{x}_j]_{j=1}^M) = [\boldsymbol{e}_j]_{j=1}^M \in \mathbb{R}^{M \times d}$. We subsequently concatenate all multi-round $L$ in each single instruction tuning to define $\boldsymbol{Z}_c = \frac{1}{L} \sum_{l=1}^L \mathbb{E}(\boldsymbol{Z}_c^l)$ which collects average text embedding of original answers ($c = a$) and their longer-context extended versions ($c = ae$) respectively. Though we adapt simple average pooling feature mechanisms, it remains an effective approach with a clear observed margin of separation between the distinct distributions (Table 6 in the Ablation study).

Given a batch size of $B$ instruction-tuning samples, we now construct three graphs $\mathcal{G}_v = (\mathcal{V}_v, \mathcal{E}_v)$, $\mathcal{G}_a = (\mathcal{V}_a, \mathcal{E}_a)$, and $\mathcal{G}_{ae} = (\mathcal{V}_{ae}, \mathcal{E}_{ae})$ representing for visual image features, text embedding encoded by LLM for original answers and their extended context embedding extended by GPT. Specifically, for each triplet pair $\{\boldsymbol{X}_v^{(k)}, [\boldsymbol{X}_a^l]^{(k)}, [\boldsymbol{X}_{ae}^l]^{(k)}\}_k, (k \in [B])$, we add a node representing $\boldsymbol{X}_v^{(k)}$ to $\mathcal{V}_v$, a node for $[\boldsymbol{X}_a^l]^{(k)}$ to $\mathcal{V}_e$, and finally a node for $[\boldsymbol{X}_{ae}^l]^{(k)}$ to $\mathcal{V}_{ae}$. This results in a set of nodes $\mathcal{V}_v = \{\boldsymbol{X}_v^{(1)}, ..., \boldsymbol{X}_v^{(B)}\}; \mathcal{V}_c = \{[\boldsymbol{X}_c^l]^{(1)}, ..., [\boldsymbol{X}_c^l]^{(B)}\}$ for each $c \in \{a, ae\}$. We equip node-level feature matrices for these graphs using their embedding computed above, i.e., $\boldsymbol{F}_v = \{\boldsymbol{Z}_v^{(1)}, ..., \boldsymbol{Z}_v^{(B)}\}, \boldsymbol{F}_c = \{\boldsymbol{Z}_c^{(1)}, ..., \boldsymbol{Z}_c^{(B)}\}$. The edges for $\mathcal{E}_v, \mathcal{E}_c$ afterward can be created through the k-nearest neighbors algorithm given the feature node matrices $\boldsymbol{F}_v, \boldsymbol{F}_c$. Finally, we can run a message-passing network $m_\alpha(.)$ on three built graphs to learn richer node representations.

This approach has proven effective for representation learning [91, 42], resulting in aggregated feature-node matrices as $\{\hat{\boldsymbol{Z}}_s^{(1)}, ..., \hat{\boldsymbol{Z}}_s^{(B)}\} = m_\alpha(\boldsymbol{F}_s, \mathcal{E}_s)$, with $s \in \{v, a, ae\}$.

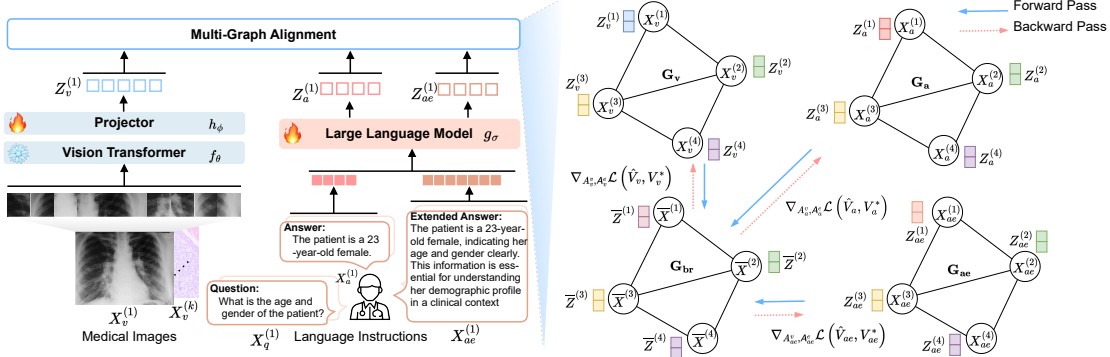

Figure 3: Overview of EXGRA-MED: The large language model $g_\sigma$ and the projector $h_\phi$ are trained jointly by aligning a triplet of modalities - input image, instruction-following data, and extended captions - through a structure-aware multigraph alignment (Eq.(3)). This alignment operates over graphs $\mathcal{G}_v$, $\mathcal{G}a$, and $\mathcal{G}ae$, representing the visual, instruction, and extended textual information, respectively, via a shared barycenter graph. The entire model is optimized end-to-end using modern black-box gradient estimation techniques to enable efficient learning across modalities [70, 63].

### 3.3 Second-order graph alignment problem

We first provide background information about the second-order graph alignment problem between two arbitrary graphs $\mathcal{G}_1 = (\mathcal{V}_1, \mathcal{E}_1)$ and $\mathcal{G}_2 = (\mathcal{V}_2, \mathcal{E}_2)$, which is known as quadratic assignment problem. It occurs in several problems in vision and computer graphics to find correspondences between two graph structures under constraints on the *consistency of node features and the graph structures* [101, 29, 25].

We denote by $\boldsymbol{V} \in \{0,1\}^{|\mathcal{V}_1||\mathcal{V}_2|}$, with $|\mathcal{V}_1| = M$ and $|\mathcal{V}_2| = N$, the indicator matrix of matched vertices, that is, $V_{i,j} = 1$ if a vertex $v_i \in \mathcal{V}_1$ is matched with $v_j \in \mathcal{V}_2$ and $V_{i,j} = 0$ otherwise. That is, $\boldsymbol{V}$ is a binary matrix with exactly one non-zero entry in each row and column. Similarly, we set $\mathbf{E} \in \{0,1\}^{|\mathcal{E}_1||\mathcal{E}_2|}$ as the indicator tensor of match edges, that is, $E_{i,k,j,l} = 1$ if $V_{i,j} = 1$ and $V_{k,l} = 1$ and $E_{i,k,j,l} = 0$ otherwise. This implies that the tensor $\mathbf{E}$ is fully determined by the matrix $\boldsymbol{V}$, that is, $E_{i,k,j,l} = V_{i,j}V_{k,l}$. We also define the vertex affinity matrix and edge affinity tensor as $\boldsymbol{A}^v \in \mathbb{R}^{|\mathcal{V}_1||\mathcal{V}_2|}$ and $\mathbf{A}^e \in \mathbb{R}^{|\mathcal{E}_1||\mathcal{E}_2|}$, respectively. The set $\mathcal{A}(\mathcal{G}_1, \mathcal{G}_2)$ indicates for all admissible pairs $(\boldsymbol{V}, \mathbf{E})$ that encode a valid matching between $\mathcal{G}_1$ and $\mathcal{G}_2$.

$$\mathcal{A}(\mathcal{G}_1, \mathcal{G}_2) = \left\{ \boldsymbol{V} \in \{0,1\}^{M \times N} : \sum_{i=1}^{M} V_{i,j} = 1, \sum_{j=1}^{N} V_{i,j} = 1 \right\}. \tag{2}$$

The second-order graph alignment (SGA) problem now is defined as:

$$\text{SGA}(\boldsymbol{A}^v, \mathbf{A}^e) = \underset{\boldsymbol{V} \in \mathcal{A}(\mathcal{G}_1, \mathcal{G}_2)}{\arg\min} \ \langle \boldsymbol{A}^v + \mathbf{A}^e \otimes \boldsymbol{V}, \boldsymbol{V} \rangle$$

$$= \underset{\boldsymbol{V} \in \mathcal{A}(\mathcal{G}_1, \mathcal{G}_2)}{\arg\min} \sum_{i,j} A_{i,j}^v V_{i,j} + \sum_{i,j,k,l} A_{i,j,k,l}^e V_{i,j} V_{k,l}. \tag{3}$$

### 3.4 Scalable Multi-graph Alignment

Our aim is to solve the graph alignment between $\mathcal{G}_v$, $\mathcal{G}_a$, and $\mathcal{G}_{ae}$ to form a triplet constraint between input image embedding, its original instruction embedding, and the extended version ones. However, solving a structure-aware graph alignment between $K$ domains ($K \geq 3$) is computationally expensive. One potential solution is to perform pairwise graph alignments $\binom{K}{2}$ times, as shown in Eq. (3), while applying specific constraints to maintain consistency between correspondences [13, 89] (Table 6). However, these approaches become impractical as $K$ increases, making them unsuitable for large-scale problems or the integration of multi-modal inputs such as text, images, audio, or medical health records.

Another direction leverages the barycenter concept from optimal transport [27, 10], which identifies a central distribution that minimizes the weighted sum of Wasserstein distances to the given input

distributions. We follow this idea to reformulate the alignment of $K$ graphs into $K$ separate alignments with a barycenter graph. Unlike previous unsupervised methods that estimate the barycenter before aligning, we directly define the barycenter using known triplet pairs across the three graphs. This significantly reduces complexity, making our solver more efficient in LLM settings.

Specifically, we define a new barycenter graph $\mathcal{G}_{br} = (\mathcal{V}_{br}, \mathcal{E}_{br})$ where $\mathcal{V}_{br} = \left\{ v_{br}^{(1)}, ..., v_{br}^{(B)} \right\}$ with $v_{br}^{(k)} = \overline{\boldsymbol{X}}^{(k)} = \{ \boldsymbol{X}_v^{(k)}, [\boldsymbol{X}_a^l]^{(k)}, [\boldsymbol{X}_{ae}^l]^{(k)} \}$ and a correspondence feature node as $\boldsymbol{F}_{br} = \frac{1}{3} \left\{ \sum_s \hat{\boldsymbol{Z}}_s^{(1)}, ..., \sum_s \hat{\boldsymbol{Z}}_s^{(B)} \right\}$ with $s \in \{v, a, ae\}$. The edge set $\mathcal{E}_{br}$ is formed similarly to another graph by running the k-nearest neighbor on feature node $\boldsymbol{F}_{br}$. We now state the multi-graph alignment as:

$$\text{SGA}(\boldsymbol{A}_s^v, \mathbf{A}_s^e) = \underset{\boldsymbol{V}_s \in \mathcal{A}(\mathcal{G}_s, \mathcal{G}_{br})}{\arg\min} \sum_{s \in \{v, a, ae\}} \langle \boldsymbol{A}_s^v + \mathbf{A}_s^e \otimes \boldsymbol{V}_s, \boldsymbol{V}_s \rangle, \tag{4}$$

where $\boldsymbol{V}_s$ is the indicator matrix representing for valid mapping between $\mathcal{G}_s$ and $\mathcal{G}_{br}$, $\boldsymbol{A}_s^v \in \mathbb{R}^{|\mathcal{V}_s||\mathcal{V}_{br}|}$ and $\mathbf{A}_s^e \in \mathbb{R}^{|\mathcal{E}_s||\mathcal{E}_{br}|}$ be vertex affinity matrix and edge affinity tensor between $\mathcal{G}_s$ and $\mathcal{G}_{br}$. For e.g., $(\boldsymbol{A}_s^v)_{ij} = d\left( \hat{\boldsymbol{Z}}_s^{(i)}, \frac{1}{3} \sum_s \hat{\boldsymbol{Z}}_s^{(j)} \right)$ with $d(.)$ be a distance metric (e.g., cosine distance) measuring similarity between node $i^{th}$ in $\mathcal{G}_s$ and node $j^{th}$ in $\mathcal{G}_{br}$.

To address the NP-Hard nature of aligning each graph to the barycenter graph $\mathcal{G}_c$, which arises from its combinatorial complexity, we employ efficient heuristic solvers utilizing Lagrange decomposition techniques [90, 80].

### 3.5 Backpropagation with Black-box Gradient Estimation

Given $\hat{\boldsymbol{V}}_s = \text{SGA}(\boldsymbol{A}_s^v, \mathbf{A}_s^e)$ be solution obtained from the solver, we aim to learn feature representation for LLMs such that $\hat{\boldsymbol{V}}_s$ be identical to true triplet alignments explicitly indicated by the barycenter graph. By denoting $\boldsymbol{V}_s^*$ be an optimal mapping between the graph $\mathcal{G}_c$ to $\mathcal{G}_{br}$, we compute the following total of hamming loss function:

$$\mathcal{L}(\hat{\boldsymbol{V}}_s, \boldsymbol{V}_s^*) = \sum_{s \in \{v, a, ae\}} \langle \hat{\boldsymbol{V}}_s, (1 - \boldsymbol{V}_s^*) \rangle + \langle \boldsymbol{V}_s^*, (1 - \hat{\boldsymbol{V}}_s) \rangle. \tag{5}$$

However, computing the gradient of the loss function with respect to the matching problem inputs $(\boldsymbol{A}_s^v, \mathbf{A}_s^e)$, i.e., $\nabla_{\boldsymbol{A}_s^v, \mathbf{A}_s^e} \mathcal{L}\left( \hat{\boldsymbol{V}}_s, \boldsymbol{V}_s^* \right)$, poses a challenge due to the piecewise constant nature of the graph matching objective in Eq. (4) [76, 79]. To address this, we resort to the IMLE [70, 63], a method that estimates gradients and enables backpropagation through the alignment algorithm by taking the difference between solutions of noise-perturbed alignments.

In particular, given $(\epsilon, \epsilon') \sim \text{Gumble}(0, 1)$ and for each $s \in \{v, a, ae\}$, we compute:

$$\left( \frac{\partial \mathcal{L}}{\partial \boldsymbol{A}_s^v}, \frac{\partial \mathcal{L}}{\partial \mathbf{A}_s^e} \right) \approx \tilde{\boldsymbol{V}}_s - \text{SGA}\left( \boldsymbol{A}_{s,\lambda}^v, \mathbf{A}_{s,\lambda}^e \right) \text{ where } \tilde{\boldsymbol{V}}_s = \text{SGA}\left( \boldsymbol{A}_s^v + \epsilon, \mathbf{A}_s^e + \epsilon' \right),$$

$$\left( \boldsymbol{A}_{s,\lambda}^v, \mathbf{A}_{s,\lambda}^e \right) = \left( \boldsymbol{A}_s^v + \epsilon, \mathbf{A}_s^e + \epsilon' \right) - \lambda \nabla_{\tilde{\boldsymbol{V}}_s} \mathcal{L}(\tilde{\boldsymbol{V}}_s, \boldsymbol{V}_s^*), \text{ with } \lambda \text{ is a step size.}$$

### 3.6 Graph Distance Properties via Structure Alignment

We provide theoretical insights into the graph-matching problem in Eq.(3). Once the optimal matching is found, it defines a valid distance metric between graphs (Theorem 1) based on joint node-edge representations. Additionally, the geodesic path connecting two graphs (Theorem 2) can be derived from these alignments, enabling dynamic formulations in optimal transport and sampling strategies in representation learning. In particular, we define a discrete distance between two graphs given a solution to the matching alignment as follows:

$$d_{\text{SGA}}(\mathcal{G}_1, \mathcal{G}_2) = \min_{\boldsymbol{V} \in \mathcal{A}(\mathcal{G}_1, \mathcal{G}_2)} \left( \sum_{i,j} A_{i,j}^v V_{i,j} + \sum_{i,j,k,l} A_{i,j,k,l}^e V_{i,j} V_{k,l} \right). \tag{6}$$

We provide an informal definition of the space of structured graphs below to introduce the main theorems. A more formal treatment, including proofs and definitions, is presented in the Appendix.

**Definition 1 (Space of all structured graphs).** $\mathbb{S}(\mathcal{F})$ *is the space of all structured graphs defined over a node metric feature space* $(\mathcal{F}, d_f)$*, where each graph is associated with an edge structure space* $(\mathcal{S}, d_s)$ *and a mixing measure* $\mu = \sum_{i=1}^{N} w_i \delta_{(f_i, s_i)}$ *over the product space* $(\mathcal{F} \times \mathcal{S})$.

---

**Theorem 1 (Metric properties).** *The distance* $d_{SGA}$ *in Eq. (6) defines a metric in* $\mathbb{S}(\mathcal{F})$.

---

The $d_{\text{SGA}}$ distance above is zero if and only if there exists a one-to-one mapping between the graph vertices that preserves both shortest paths and features and both graphs have the same number of vertices.

---

**Theorem 2 (Geodesic space).** *The space* $\mathbb{S}(\mathcal{F})$ *equipped with the* $d_{SGA}$ *distance is geodesic.*

---

# 4 Experiments

## 4.1 Implementation Details

Table 1: Fine-tuning performance on MedVQA tasks (**pre-trained 10%**). **Bold** indicates the best values among pre-training algorithms (Sec. 4.3) , excluding LLaVA-Med.

| Method | VQA-RAD | | | SLAKE | | | PathVQA | | | Overall |
|---|---|---|---|---|---|---|---|---|---|---|
| | Open | Closed | Avg. | Open | Closed | Avg. | Open | Closed | Avg. | |
| LLaVA-Med (100%) | 63.65 | 81.62 | 72.64 | 83.44 | 83.41 | 83.43 | 36.78 | 91.33 | 64.06 | 73.37 |
| LLaVA-Med (10%) | 43.38↓20.3 | 61.4↓20.2 | 52.39↓20.3 | 80.94↓2.5 | 80.29↓3.1 | 80.62↓2.8 | 24.26↓13.69 | 88.03↓3.18 | 56.15↓7.91 | 63.05↓10.3 |
| InfoNCE | 59.39 | 77.57 | 68.48 | 82.4 | 83.17 | 82.78 | 34.59 | 91.45 | 63.02 | 71.43 |
| PLOT | 16.86 | 26.47 | 21.67 | 37.81 | 56.25 | 47.03 | 11.79 | 81.36 | 46.58 | 38.42 |
| SigLIP | 56.99 | 77.94 | 67.47 | 80.86 | 80.53 | 80.69 | 18.08 | 50.85 | 34.465 | 60.88 |
| VLAP | 57.49 | 76.47 | 66.98 | 80.05 | 82.21 | 81.13 | 32.21 | 91.16 | 61.685 | 69.93 |
| GeoCLAP | 60.68 | 75.37 | 68.03 | 82.64 | **85.10** | 83.87 | 35.12 | 91.15 | 63.14 | 71.68 |
| PAC-S | 57.72 | 72.79 | 65.26 | 83.78 | 81.49 | 82.64 | 35.01 | 91.36 | 63.19 | 70.36 |
| IMAGEBIND | 57.31 | 75.74 | 66.53 | 80.79 | 84.13 | 82.46 | 34.61 | 91.42 | 63.02 | 70.67 |
| **ExGra-Med** | **66.02** | **79.04** | **72.52** | **84.92** | **85.10** | **85.01** | **37.25** | **91.45** | **64.34** | **73.96** |

Table 2: Fine-tuning performance on MedVQA tasks (**pre-trained 40%**). **Bold** indicate for best values among pre-training algorithms (Sec. 4.3) excluding LLaVA-Med.

| Method | VQA-RAD | | | SLAKE | | | PathVQA | | | Overall |
|---|---|---|---|---|---|---|---|---|---|---|
| | Open | Closed | Avg. | Open | Closed | Avg. | Open | Closed | Avg. | |
| LLaVA-Med (100%) | 63.65 | 81.62 | 72.64 | 83.44 | 83.41 | 83.43 | 36.78 | 91.33 | 64.06 | 73.37 |
| LLaVA-Med (40%) | 62.23↓1.4 | 79.41↓2.2 | 70.82↓1.8 | 84.42↑1.0 | 83.65↓0.2 | 84.04↑0.6 | 31.86↓4.9 | 84.99↓6.3 | 58.43↓5.6 | 71.09↓2.3 |
| InfoNCE | 63.11 | 77.57 | 70.34 | 82.68 | 83.89 | 83.29 | 33.58 | 89.62 | 61.6 | 71.74 |
| PLOT | 64.36 | 79.41 | 71.89 | 83.38 | 82.93 | 83.16 | 35.11 | 89.59 | 62.35 | 72.46 |
| SigLIP | 63.02 | 81.25 | 72.14 | 81.26 | 80.29 | 80.77 | 36.01 | 90.86 | 63.435 | 72.12 |
| VLAP | 63.17 | 79.04 | 71.11 | 83.38 | 83.89 | 83.64 | 35.62 | 90.83 | 63.225 | 72.66 |
| GeoCLAP | 62.28 | **82.72** | 72.5 | 82.64 | 85.2 | 83.92 | 33.2 | 75.05 | 54.13 | 70.18 |
| PAC-S | 63.77 | 79.41 | 71.59 | **84.52** | 85.58 | 85.05 | 27.11 | 85.34 | 56.23 | 70.96 |
| IMAGEBIND | 64.73 | 78.68 | 71.71 | 82.31 | 84.62 | 83.47 | 35.76 | 87.08 | 61.42 | 72.20 |
| **ExGra-Med** | **66.01** | **82.72** | **74.37** | 84.47 | **85.82** | **85.15** | **37.41** | **91.27** | **64.34** | **74.57** |

Table 3: ExGra-Med versus LLaVa-Med when pre-trained with extended captions (ext.cap) . Performance is reported on VQA-RAD downstream after fine-tuning.

| Method | VQA-RAD | | |
|---|---|---|---|
| | Open | Closed | Avg. |
| LLaVA-Med (100%) | 63.65 | 81.62 | 72.64 |
| LLaVA-Med (10%) | 43.38↓20.3 | 61.4↓20.2 | 52.39↓20.3 |
| LLaVA-Med (10%) ext.cap | 63.07 | 75.74 | 69.41 |
| **ExGra-Med (10%)** | **66.02** | **79.04** | **72.52** |
| LLaVA-Med (40%) | 62.23↓1.4 | 79.41↓2.2 | 70.82↓1.8 |
| LLaVA-Med (40%) ext.cap | 63.68 | 79.78 | 71.73 |
| **ExGra-Med (40%)** | **66.01** | **82.72** | **74.37** |

Table 4: EXGRA-MED results with different frozen LLMs . It shows that Gemini and Qwen is also effective within our method.

| Method | VQA-RAD | SLAKE |
|---|---|---|
| ExGra-Med (GPT-4), 10% | **72.52** | **85.01** |
| ExGra-Med (Gemini), 10% | 71.09 | 83.98 |
| ExGra-Med (Qwen), 10% | 70.13 | 84.7 |
| LLaVA-Med (Baseline) 10% | 52.39 | 80.62 |
| ExGra-Med (GPT-4), 40% | **74.37** | **85.15** |
| ExGra-Med (Gemini), 40% | 73.26 | 85.10 |
| ExGra-Med with synonyms, 40% | 72.39 | 82.93 |
| LLaVA-Med (Baseline) 40% | 70.82 | 84.04 |

**Model Architecture Configuration.** We use the LLaMA-7B large language model [93], the CLIP-ViT-L-Patch14 visual encoder [77], and an MLP projection similar to LLaVA 1.5 [56]. *Stage 1* follows the standard LLaVA-Med [50] setup, while *stage 2 incorporates our multi-graph alignment with autoregressive training*. For multi-graph alignment, a 2-layer graph convolutional network is applied to the output of the Projection and LLM Decoder (handling both image and text modalities). We train for 1 epoch in stage 1 and 3 epochs in stage 2 using the same dataset as LLava-Med. The model is optimized using Adam [47] with CosineAnnealingLR scheduler and learning rates of $2e-3$ and $2e-5$ for stages 1 and 2, respectively.

**Pre-training data.** We use the same dataset as LLaVA-Med [50]. Stage 1 includes 600K image-text pairs filtered from PMC-15M, converted into instruction-following data with simple image descriptions. Stage 2 comprises 60K image-text pairs from PMC-15M across five modalities: CXR, CT, MRI, histopathology, and gross pathology. GPT-4 is then employed to generate multi-round Q&A in a tone mimicking visual interpretation, converting these pairs into an instruction-following format.

**Running-time.** We train EXGRA-MED using 4 A100-GPUs per with 80GB for both stages and complete the training process for stage 1 in 6.5 hours and for stage 2 in 7.5 hours. With original LLaVA-Med (version 1.5) [50], the training process for stage 1 finishes in 6.5 hours, and for stage 2 finishes in 7 hours. In total, we need an extra 0.5 hour to complete the whole pre-training process compared to the LLaVa-Med.

## 4.2 Autoregressive Training is Data-hungry

We highlight the data-intensive nature of autoregressive training by evaluating LLaVA-Med, a state-of-the-art biomedical multimodal language model. LLaVA-Med follows a two-stage training process: Stage 1 aligns image-text tokens with biomedical concepts, and Stage 2 fine-tunes the model for instruction tasks. We pre-trained LLaVA-Med on varying data amounts (10%, 40%, 70%) and fine-tuned it on the VQA-RAD dataset. As shown in Figure 1, performance drops significantly at 10% pre-training compared to full training, revealing the autoregressive mechanism's data dependency issue in medical-MLLMs. This highlights the challenge of weak connections between visual features and text embeddings without sufficient instruction-tuning data.

In contrast, our EXGRA-MED specifically learns image-text alignment by enforcing semantic consistency across images, instruction responses, and extended contexts. Using the same setup as LLaVA-Med, EXGRA-MED excels even with limited data, achieving $72.52\%$ at just $10\%$ pretraining, far above LLaVA-Med's $52.39\%$ and consistently outperforms LLaVA-Med across instruction tuning rates ($40\% - 100\%$).

**Impact of Longer Training and Enriched Captions on LLAVA-MED Performance**. We conduct a deeper analysis of the data-hungry phenomenon by examining LLAVA-MED when (i) *trained longer in Stage 2* (with an additional hour) and when (ii) *incorporating extended captions* in an autoregressive manner, as done in EXGRA-MED. The experiments were performed using 10% and 40% pre-training settings, followed by fine-tuning on the VQA-RAD dataset. From the results in Table 3, we draw two key conclusions. First, extended captions improve LLAVA-MED performance, particularly in the data-scarce 10% pre-training scenario. However, in both settings, EXGRA-MED demonstrates superior performance with significant margins, *highlighting the effectiveness of its multi-graph alignment strategy* in mitigating the data-hungry issues of the autoregressive mechanism.

## 4.3 Multi-modal Pre-training Comparison

To further demonstrate the strengths of EXGRA-MED, we compare its performance against other vision-language pre-training methods currently used for visual instruction tuning to enhance frozen vision-language models

**Datasets.** We test pre-trained models on three prominent biomedical VQA datasets: VQA-RAD [48], SLAKE [55], and PathVQA [31]. VQA-RAD includes 3,515 questions across 315 radiology images, while SLAKE contains 642 radiology images from various body parts and over 7k QA pairs. PathVQA, focused on pathology, features 5k images and 32.8k questions. All datasets include **open-ended** (e.g., what, why, where) and **closed-ended** (yes/no or two-option) question types. We report performance using recall for open-ended, which evaluates the proportion of ground-truth tokens that appear in the generated sequences and accuracy for the closed-ended questions.

**Baselines.** We compare *seven approaches*, including:

- **two modalities-based methods** such as InfoNCE-based methods [45, 58], SigLIP [102], PLOT [15], VLAP [72]. Among this, SigLIP adapts the Sigmoid loss on image-text pairs to break the global view of the pairwise similarities for normalization, resulting in scaling in large batch sizes. PLOT defines optimal transport as a distance between visual image patches and text embedding. In contrast, VLAP uses assignment prediction to bridge the modality gap between the visual and LLM embeddings.

- **multi-modal learning across three modalities** such as image, text, voice, or augmented image—is explored by methods like PAC-S [81], GeoCLAP [46], and IMAGEBIND [26]. PAC-S integrates contrastive losses across multiple modality pairs: (image–text), (image–augmented text), and (text–augmented text). GeoCLAP applies CLIP-style contrastive learning to all cross-domain pairs, such as audio–text and text–image. Similarly, IMAGEBIND generalizes the InfoNCE loss to learn unified embeddings across diverse modalities.

We train the baselines under the same settings as EXGRA-MED with varying pre-training data rates and compare their performance on downstream tasks.

**Results.** Tables 1–16 show our performance and baselines under $10\%$, $40\%$, and $100\%$ instruction-tuning data. While most contrastive baselines improve over LLaVA-Med at $10\%$, EXGRA-MED consistently outperforms all methods across settings. It excels particularly on open-ended questions requiring external knowledge and maintains stable gains across all VQA datasets. In contrast, some

Table 5: Comparison with other Med-MLLMs on MedVQA tasks. *All models (except GPT-4) are fine-tuned on the same training set in each VQA task*. These Med-MLLMs differ notably in model size, training data volume, and pre-training strategies - e.g., ExGra-Med (7B, 60K GPT-4 augmented samples) vs. MedDR (40B, 2M samples).

| Method | #Para | VQA-RAD | | | SLAKE | | | PathVQA | | | Overall |
|---|---|---|---|---|---|---|---|---|---|---|---|
| | | Open | Closed | Avg. | Open | Closed | Avg. | Open | Closed | Avg. | |
| LLaVA-Med | 7B | 63.65 | 81.62 | 72.64 | 83.44 | 83.41 | 83.43 | 36.78 | 91.33 | 64.06 | 73.37 |
| BiomedGPT-B | 182M | 60.9 | 81.3 | 71.1 | 84.3 | 89.9 | 87.1 | 28 | 88 | 58 | 72.07 |
| M2I2 | - | 61.8 | 81.6 | 71.7 | 74.7 | 91.1 | 82.9 | 36.3 | 88 | 62.15 | 72.25 |
| BioMed-CLIP | 422M | 67.6 | 79.8 | 73.7 | 82.5 | 89.7 | 86.1 | | | | |
| Med-Dr | 40B | 37.5 | 78.9 | 58.2 | 74.2 | 83.4 | 78.8 | 33.5 | 90.2 | 61.85 | 66.28 |
| LLaVA (general) | 7B | 50 | 65.1 | 57.55 | 78.2 | 63.2 | 70.7 | 7.7 | 63.2 | 35.45 | 54.57 |
| GPT-4 | 200B | 39.5 | 78.9 | 59.2 | 33.6 | 43.6 | 38.6 | | | | |
| Med-MoE (Phi2) | 3.6B | 58.55 | 82.72 | 70.64 | 85.06 | 85.58 | 85.32 | 34.74 | 91.98 | 63.36 | 73.11 |
| Med-MoE (Stable LM) | 2B | 50.08 | 80.07 | 65.08 | 83.16 | 83.41 | 83.29 | 33.79 | 91.30 | 62.55 | 70.3 |
| **ExGra-Med** | 7B | 66.35 | **83.46** | 74.91 | **85.34** | 85.58 | 85.46 | 36.82 | 90.92 | 63.87 | 74.75 |
| **ExGra-Med (DCI)** | 7B | 67.03 | **83.46** | 75.25 | 84.88 | 85.58 | 85.23 | **37.77** | 91.86 | 64.82 | 75.1 |

Table 6: ExGra-Med ablation study. Results are presented as average scores on VQA-RAD and SLAKE, using pre-trained weights on 10%, 40%, 100%.

| Method | VQA-RAD | SLAKE |
|---|---|---|
| ExGra-Med (Full, 10%, $\alpha = 1.0$) | 72.52 | 85.01 |
| - (ii) ExGra-Med (Full, 10%, $\alpha = 0.1$) | 65.95 | 82.9 |
| - (ii) ExGra-Med (Full, 10%, $\alpha = 0.5$) | 67.72 | 82.33 |
| ExGra-Med (Full, 40%) | 74.37 | 84.99 |
| - (iii) ExGra-Med w/o ext. context | 72.12 | 81.95 |
| - (iv) ExGra-Med w/o ori. caption | 72.58 | 82.31 |
| - (v) ExGra-Med w/o message passing | 73.90 | 84.29 |
| - (vi) ExGra-Med in two stages | 72.81 | 84.14 |
| ExGra-Med (Full, 100%) | 74.91 | 85.46 |
| - (vii) ExGra-Med w/o barycenter graph | 73.88 | 84.34 |

methods like `SigLIP` peak at 40% (e.g., 72.14% Avg on VQA-RAD) but drop notably at 100% (down over 6%), whereas EXGRA-MED improves further to 74.91% (Avg) and 74.75% (Overall).

## 4.4 Med-VQA Comparison with Medical MLLMs

We now compare EXGRA-MED pre-trained with 100% data against other medical foundation models, each trained on varying datasets and employing different architectures or model sizes.

**Baselines.** We benchmark *eight competitors*, both generic or medical foundation models, including `LLaVA` [57], `LLaVA-Med` [50], `Med-Flamingo` [64], `Med-Dr` [30], `Biomed-GPT` [104], `M2I2` [53], `GPT-4o` [7] and `Med-MoE` [39]. Whilst `LLaVA` and `GPT-4o` have no medical background, the others are pre-trained on a variety of biomedical knowledge. With the exception of `LLaVa`, which we reproduced, the results for the other baselines are taken from the literature. Moreover, we also present an enhanced version, EXGRA-MED + DCI, which integrates multi-scale visual features from vision encoders [100], potentially benefiting medical image analysis by considering both local (detailed) and global (contextual) features.

**Results.** Overall, both versions of EXGRA-MED outperform baseline models (Table 5), with the DCI variant achieving the best results on PathVQA (64.82% average, 75.1% overall). Compared to `LLaVA-Med`, it shows notable gains: +2.01% on VQA-RAD, +2.03% on SLAKE, and +0.76% on PathVQA. Despite using only 7B parameters, both EXGRA-MED models surpass the 40B `Med-Dr` across all datasets.

## 4.5 Medical Visual Chatbot Evaluation & Zero-shot Image Classification

**Medical Visual Chatbot.** We present in Section E Appendix EXGRA-MED results compared against several SOTA general and Med-MLLMs such as LLaVA, GPT-4o, LLaVA-Med, Med-Flamingo, Med-Dr, and Biomed-GPT. Among these, we observe that EXGRA-MED is the best model across question types.

**Zero-shot Image Classification as MedVQA.** Results are presented in Section F of the Appendix. In short, we outperform other models across all datasets, particularly excelling in microscopy, where it surpasses RadFM by 8.2%

## 4.6 Additional Analysis

Table 7: Results of fully finetuning vs LoRA finetuning on VQA-RAD dataset.

| Method | VQA-RAD Open | VQA-RAD Closed | VQA-RAD Avg | Param |
|---|---|---|---|---|
| LLaVa-Med (LoRa) | 62.06 | 75.00 | 68.53 | 2.1B |
| ExGra-Med (LoRa) | 63.55 | 79.41 | 71.48 | 2.1B |
| LLaVa-Med (Fully-finetuning) | 63.65 | 81.62 | 72.64 | 7B |
| ExGra-Med (Fully-finetuning) | 66.35 | 83.46 | 74.91 | 7B |

**Potentially Hallucination in Extended Captions.** We conducted a user study with *five general practitioners* from top public hospitals (Appendix Section K). In Stage 2 of pre-training, each expert evaluated 200 image-text pairs (1,000 total) across five modalities - chest X-ray, CT, MRI, histology, and others - rating the completeness and accuracy of GPT-4-generated extended captions. As shown

in Figures 10–14, most scores ranged from 3 to 5, with few low ratings, confirming the overall consistency and quality of the extended outputs. Also, these extended captions are used only during pre-training to guide latent space alignment. *They are excluded during fine-tuning* on downstream tasks. As such, we argue that a small amount of noise in the extended captions should have minimal impact on overall performance, since they do not directly affect the model's task-specific adaptations.

**Other Factors.** We examine ExGra-Med results under following settings:

- (i) *generalization to other frozen LLMs* (GPT-4, Gemini [92], and Qwen3-8B LLM [99]) to generate extended captions and how the method works with *simple synonyms* (Table 4).
- ii) *contribution of coefficient ($\alpha$) combine multi-graph alignment with auto-regressive*.
- iii) *without using extended contexts*, i.e., simplifying from the three graph alignment to two cross-domain alignments (image vs. original captions).
- (iv) *without using original captions*, i.e., only extended ones are used.
- (v) applying *message passing to enhance node features*.
- (vi) *using multi-graph alignment* in both steps (default uses only Step-2).
- (vii) *solving three pair-wise graph alignments* in multi-graph alignment *rather than solving through a barycenter graph* (Eq.(4) in Sec. 3.4).
- (viii) using *parameter-efficient finetuning* with LoRa [33] rather than fully finetuning on downstream tasks.

Tables 4-6 show results (i)-(vii) where the most important factors are highlighted. We further observe that ExGra-Med generalizes effectively across distinct LLM paraphrase generators such as GPT-4, Gemini, and Qwen3-8B. The stable performance across these models indicates that ExGra-Med is not tightly coupled to a specific language model architecture, but instead captures transferable alignment mechanisms applicable to a wide range of paraphrastic contexts.

In Table 7, we report the results of ExGra-Med and LLaVA-Med on the VQA-RAD dataset for the (viii) setting. As shown, ExGra-Med consistently outperforms LLaVA-Med even when both models are fine-tuned using LoRA, demonstrating the robustness of our approach under parameter-efficient adaptation. Nevertheless, as expected, LoRA-based fine-tuning yields a modest performance drop compared to full fine-tuning. We believe that to bridge this gap, pre-training on a larger-scale medical instruction-tuning dataset (e.g., the 21M samples curated from MedTrinity [98]) would allow the LoRA setup to more closely match the performance of fully fine-tuned models. Additional analyses on average pooling features, k-nearest neighbors for graph construction (Section 3.2).

**Current Limitations and Future Work.** While our experiments have primarily focused on the LLaVa model, it is essential to validate the effectiveness and adaptability of EXGRA-MED with alternative architectures, such as the Flamingo model [9], which has shown promising results in vision-language tasks. Expanding the evaluation to include other state-of-the-art models can provide a broader perspective on the generalizability and robustness of EXGRA-MED. Furthermore, incorporating vision encoders or large language models (LLMs) that are specifically pre-trained on medical datasets [19, 62, 107, 17, 103] presents a compelling opportunity to enhance both performance and domain-specific understanding. These specialized models are designed to capture the nuanced characteristics of medical data, which could further enhance the robustness of EXGRA-MED in complex biomedical scenarios. Moreover, extending our mechanism to the setting of medical visual chain-of-thought [41, 49] reasoning represents a promising direction for improving both the overall performance and the trustworthiness of the model.

## 5   Conclusion

We have shown that enforcing triplet correlations among images, instructions, and extended captions significantly enhances vision-language alignment - an area where autoregressive models like LLAVA-MED struggle, particularly under limited data and domain shift, which manifests as a strong dependence on large-scale pre-training data. To this end, we introduce EXGRA-MED, a multi-graph alignment algorithm that trains efficiently, matches LLAVA-MED using only 10% of the data, and outperforms other state-of-the-art models across tasks. These results highlight that selecting an effective learning algorithm for LLMs is as crucial as scaling model size or data volume.

## Acknowledgment

This work was supported by Deutsche Forschungsgemeinschaft (DFG, German Research Foundation) under Germany's Excellence Strategy - EXC 2075 – 390740016, the DARPA ANSR program under award FA8750-23- 2-0004, the DARPA CODORD program under award HR00112590089. The authors thank the International Max Planck Research School for Intelligent Systems (IMPRS-IS) for supporting Duy M. H. Nguyen. Duy M. H. Nguyen and Daniel Sonntag are also supported by the No-IDLE project (BMBF, 01IW23002), the MASTER project (EU, 101093079), and the Endowed Chair of Applied Artificial Intelligence, Oldenburg University.

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

SUPPLEMENTARY MATERIAL FOR
"EXGRA-MED: EXTENDED CONTEXT GRAPH ALIGNMENT FOR
MEDICAL VISION-LANGUAGE MODELS"

## Contents

# A Proofs of the Main Theoretical Results

In this section, we provide detailed technical proofs of our main theoretical results. To accomplish this, we first introduce the fundamental concepts and materials that will be utilized in the proofs of Theorems 1 and 2.

We consider labelled graphs as tuples of the form $\mathcal{G} = (\mathcal{V}, \mathcal{E}, \mathcal{L}_f, \mathcal{L}_s)$, where the labelling function $\mathcal{L}_f : \mathcal{V} \mapsto \mathcal{F}$ assigns each vertex $v_i \in \mathcal{V}$ to a feature $f_i = \mathcal{L}_f(v_i)$ in some feature space $(\mathcal{F}, d_f)$. Similarly, we denote $\mathcal{L}_s : \mathcal{V} \mapsto \mathcal{S}$ as a structure function which links each vertex $v_i \in \mathcal{V}$ with its structure information $s_i = \mathcal{L}_s(v_i)$, *e.g.*, edge information, in some structure space $(\mathcal{S}, d_s)$. By associating a weight to each vertex, we allow the graph $\mathcal{G}$ to be represented by a fully supported mixing measure $\mu = \sum_{i=1}^{N} w_i \delta_{(f_i, s_i)}$ over the product between feature space and structure space $\mathcal{F} \times \mathcal{S}$. Notably, $\mu$ is not necessarily a probability measure as the summation of its weights can be different from one. We have the vertex affinity matrix between two graphs as $\boldsymbol{A}^v \in \mathbb{R}^{M \times N}$, where $A_{i,j}^v = (d_f(f_i, f_j))i, j$. Structural similarity is measured by pairwise distances within each graph, represented by $\boldsymbol{A}^e \in \mathbb{R}^{|\mathcal{E}_1||\mathcal{E}_2|}$, with $A_{i,j,k,l}^e = |d_s(s_i, s_k) - d_s(s_j, s_l)|$, where $d_s(\cdot)$ models node distance, such as the shortest path. We then define the space of all structured graphs $(\mathcal{F} \times \mathcal{S}, d_f, \mu)$ over a metric feature space $(\mathcal{F}, d_f)$ as $\mathbb{S}(\mathcal{F})$, where $(\mathcal{S}, d_s)$ is a metric structure space and $\mu = \sum_{i=1}^{N} w_i \delta_{(f_i, s_i)}$ is a mixing measure over $\mathcal{F} \times \mathcal{S}$.

## A.1 Proof of Theorem 1

For the sake of simplicity, we denote the labeled graphs $\mathcal{G}$ and structured graphs discussed above only by $\mu$ the whole structured graph.

To prove Theorem 1, for any two graphs $\mathcal{G}_1$ and $\mathcal{G}_2$ in the structured graph space $\mathbb{S}(\mathcal{F})$, described respectively by their mixing measure $\mu_1 = \sum_{i=1}^{M} w_{1i} \delta_{(f_{1i}, s_{1i})}$ and $\mu_2 = \sum_{j=1}^{N} w_{2j} \delta_{(f_{2j}, s_{2j})}$, respectively, we wish to prove the following properties:

1. Positivity: $d_{\text{SGA}}(\mathcal{G}_1, \mathcal{G}_2) > 0$ for any $\mathcal{G}_1 \neq \mathcal{G}_2$.
2. Equality relation: $d_{\text{SGA}}(\mathcal{G}_1, \mathcal{G}_2) = 0$ if and only if $\mathcal{G}_1 = \mathcal{G}_2$.
3. Symmetry: $d_{\text{SGA}}(\mathcal{G}_1, \mathcal{G}_2) = d_{\text{SGA}}(\mathcal{G}_2, \mathcal{G}_1)$.
4. Triangle inequality: $d_{\text{SGA}}(\mathcal{G}_1, \mathcal{G}_3) \leq d_{\text{SGA}}(\mathcal{G}_1, \mathcal{G}_2) + d_{\text{SGA}}(\mathcal{G}_2, \mathcal{G}_3)$ for any graph $\mathcal{G}_3$.

Note first that 1. Positivity and 3. Symmetry hold trivially.

**Proof of 2. Equality relation.** The equality relation immediately follows the following Proposition 1, which is proved in Appendix B.1.

**Proposition 1** (Equality relation). *For any two graphs $\mathcal{G}_1$ and $\mathcal{G}_2$ in the structured graph space $\mathbb{S}(\mathcal{F})$, described respectively by their mixing measure $\mu_1 = \sum_{i=1}^{M} w_{1i} \delta_{(f_{1i}, s_{1i})}$ and $\mu_2 = \sum_{j=1}^{N} w_{2j} \delta_{(f_{2j}, s_{2j})}$, it holds $d_{SGA}(\mathcal{G}_1, \mathcal{G}_2) = 0$ if and only if $M = N$ and there exists a bijection $\sigma : [M] \mapsto [M]$ such that:*

*E1.* $\forall i \in [M] : w_{1i} = w_{2\sigma(i)}$.

*E2.* $\forall i \in [M] : f_{1i} = f_{2\sigma(i)}$.

*E3.* $\forall i, k \in [M]^2 : d_s(s_{1i}, s_{1k}) = d_s(s_{2\sigma(i)}, s_{2\sigma(k)})$.

**Proof of 4. Triangle inequality.** Let us consider two arbitrary graphs $\mathcal{G}_1$ and $\mathcal{G}_2$, described respectively by their probability measure $\mu_1 = \sum_{i=1}^{M} w_{1i} \delta_{(f_{1i}, s_{1i})}$ and $\mu_2 = \sum_{j=1}^{N} w_{2j} \delta_{(f_{2j}, s_{2j})}$. For any graph $\mathcal{G}_3$ described by its probability measure $\mu_3 = \sum_{i=1}^{K} w_{3k} \delta_{(f_{3k}, s_{3k})}$, we define $\boldsymbol{P} \in \mathcal{A}(\mathcal{G}_1, \mathcal{G}_2)$ and $\boldsymbol{Q} \in \mathcal{A}(\mathcal{G}_2, \mathcal{G}_3)$ as two optimal couplings of the SGA distance between $\mu_1$ and $\mu_2$ and $\mu_2$ and $\mu_3$, respectively, *i.e.*,

$$\boldsymbol{P} \in \mathcal{A}(\mathcal{G}_1, \mathcal{G}_2) = \left\{ \boldsymbol{P} \in \{0, 1\}^{M \times N} : \sum_{i=1}^{M} P_{i,j} = w_{1j} = 1, \sum_{j=1}^{N} P_{i,j} = w_{2i} = 1 \right\},$$

$$\boldsymbol{Q} \in \mathcal{A}(\mathcal{G}_2, \mathcal{G}_3) = \left\{ \boldsymbol{Q} \in \{0,1\}^{N \times K} : \sum_{j=1}^{N} Q_{j,k} = w_{2k} = 1, \sum_{k=1}^{K} Q_{j,k} = w_{3j} = 1 \right\}.$$

We then construct $\boldsymbol{R} = \left( \sum_j \frac{P_{i,j} Q_{j,k}}{w_{2j}} \right)_{i,k}$. Then it holds that $\boldsymbol{R} \in \mathcal{A}(\mathcal{G}_1, \mathcal{G}_3)$. Indeed, we have

$$\sum_i R_{i,k} = \sum_i \sum_j \frac{P_{i,j} Q_{j,k}}{w_{2j}} = \sum_j \sum_i P_{i,j} \frac{Q_{j,k}}{w_{2j}} = \sum_j w_{1j} \frac{Q_{j,k}}{w_{2j}} = \sum_j Q_{j,k} = 1.$$

By the suboptimality of $\boldsymbol{R}$, the triangle inequalities of $d_f$ and $|\cdot|$, we have

$$d_{\text{SGA}}(\mathcal{G}_1, \mathcal{G}_3) \leq \sum_{i,j,k,l} \left[ d_f(f_{1i}, f_{3j}) + |d_s(s_{1i}, s_{1k}) - d_s(s_{3j}, s_{3l})| \right] R_{i,j} R_{k,l}$$

$$= \sum_{i,j,k,l} \left[ d_f(f_{1i}, f_{3j}) + |d_s(s_{1i}, s_{1k}) - d_s(s_{3j}, s_{3l})| \right] \sum_t \frac{P_{i,t} Q_{t,j}}{w_{2t}} \sum_d \frac{P_{k,d} Q_{d,l}}{w_{2d}}$$

$$= \sum_{i,j,k,l,t,d} \left[ d_f(f_{1i}, f_{3j}) + |d_s(s_{1i}, s_{1k}) - d_s(s_{3j}, s_{3l})| \right] \frac{P_{i,t} Q_{t,j}}{w_{2t}} \frac{P_{k,d} Q_{d,l}}{w_{2d}}$$

$$\leq \sum_{i,j,k,l,t,d} \left[ d_f(f_{1i}, f_{2t}) + d_f(f_{2t}, f_{3j}) \right] \frac{P_{i,t} Q_{t,j}}{w_{2t}} \frac{P_{k,d} Q_{d,l}}{w_{2d}}$$

$$+ \sum_{i,j,k,l,t,d} \left[ |d_s(s_{1i}, s_{1k}) - d_s(s_{2t}, s_{2d})| + |d_s(s_{2t}, s_{2d}) - d_s(s_{3j}, s_{3l})| \right] \frac{P_{i,t} Q_{t,j}}{w_{2t}} \frac{P_{k,d} Q_{d,l}}{w_{2d}}$$

$$= \sum_{i,j,k,l,t,d} \left[ d_f(f_{1i}, f_{2t}) + |d_s(s_{1i}, s_{1k}) - d_s(s_{2t}, s_{2d})| \right] \frac{P_{i,t} P_{k,d}}{w_{2t}} \frac{Q_{t,j} Q_{d,l}}{w_{2d}}$$

$$+ \sum_{i,j,k,l,t,d} \left[ d_f(f_{2t}, f_{3j}) + |d_s(s_{2t}, s_{2d}) - d_s(s_{3j}, s_{3l})| \right] \frac{P_{i,t} Q_{t,j}}{w_{2t}} \frac{P_{k,d} Q_{d,l}}{w_{2d}}$$

$$= \sum_{i,k,t,d} \left[ d_f(f_{1i}, f_{2t}) + |d_s(s_{1i}, s_{1k}) - d_s(s_{2t}, s_{2d})| \right] P_{i,t} P_{k,d} \sum_j \frac{Q_{t,j}}{w_{2t}} \sum_l \frac{Q_{d,l}}{w_{2d}}$$

$$+ \sum_{j,l,t,d} \left[ d_f(f_{2t}, f_{3j}) + |d_s(s_{2t}, s_{2d}) - d_s(s_{3j}, s_{3l})| \right] Q_{t,j} Q_{d,l} \sum_i \frac{P_{i,t}}{w_{2t}} \sum_k \frac{P_{k,d}}{w_{2d}}.$$

Note that we have

$$\sum_j \frac{Q_{t,j}}{w_{2t}} = \sum_l \frac{Q_{d,l}}{w_{2d}} = \sum_i \frac{P_{i,t}}{w_{2t}} = \sum_k \frac{P_{k,d}}{w_{2d}} = 1.$$

This is how we achieve the desired result, because

$$d_{\text{SGA}}(\mathcal{G}_1, \mathcal{G}_3) \leq \sum_{i,k,t,d} \left[ d_f(f_{1i}, f_{2t}) + |d_s(s_{1i}, s_{1k}) - d_s(s_{2t}, s_{2d})| \right] P_{i,t} P_{k,d}$$

$$+ \sum_{j,l,t,d} \left[ d_f(f_{2t}, f_{3j}) + |d_s(s_{2t}, s_{2d}) - d_s(s_{3j}, s_{3l})| \right] Q_{t,j} Q_{d,l}$$

$$= d_{\text{SGA}}(\mathcal{G}_1, \mathcal{G}_2) + d_{\text{SGA}}(\mathcal{G}_2, \mathcal{G}_3) \text{ (since } \boldsymbol{P} \text{ and } \boldsymbol{Q} \text{ are the optimal plans).}$$

### A.2 Proof of Theorem 2

Theorem 2 enables us to characterise the optimal transport problem between two measures as a curve in the space of measures, with the objective of minimising its total length. Furthermore, this formulation is beneficial for deriving global minima results for non-convex particles in gradient descent in an optimisation context, which is a valuable application of gradient flows [20]. By definition, a geodesic between $\mathcal{G}_1$ and $\mathcal{G}_2$ is a shortest path between these two graphs. In particular, the computation of distances along constant speed geodesic paths is a relatively straightforward process, as these paths are directly embedded into the real line $\mathbb{R}$ as follows: $d_{\text{SGA}}(\mathcal{G}_1, \mathcal{G}_2) = |t - u|^{-1} d_{\text{SGA}}(p(u), p(t))$, for all $0 \leq u \neq t \leq 1$ and for any path (continuous map) $p$ connect $\mathcal{G}_1$ to

$\mathcal{G}_2$ such that $p(u) = \mathcal{G}_1$ and $p(t) = \mathcal{G}_2$. To prove Theorem A.2, it is necessary to collect fundamental material using Definition 2 from metric geometry for a general metric space $(\mathbb{M}, d)$.

**Definition 2** (Length and geodesic spaces). *Let $(\mathbb{M}, d)$ be a metric space and two points $x, y \in \mathbb{M}$. We say that a path (curve) $p : [0, 1] \mapsto \mathbb{M}$ connect or join $x$ to $y$ if $p(0) = x$ and $p(1) = y$ and $p$ is a continuous map.*

*We also define the length $L(p) \in \mathbb{R}$ of a path $p : [0, 1] \mapsto \mathbb{M}$ as*

$$L(p) := \sup \sum_{i=1}^{n} d(p(t_i), p(t_{i+1}))$$

*where we take the supremum over all $n \geq 1$ and all $n$-tuples $t_1 < \ldots < t_n$ in $[0, 1]$.*

*We denote a metric space $\mathbb{M}$ as a length space if for all $x, y \in \mathbb{M}$, $d(x, y) = \inf_p L(p)$ where the infimum is taken over all paths $p$ connecting $x$ to $y$.*

*We call a length space as a geodesic space if for all $x, y \in \mathbb{M}$, there exists a path $p(x, y) : [0, 1] \mapsto \mathbb{M}$ such that*

$$d(x, y) = \min_{p(x,y)} L(p(x, y)).$$

*We also denote the path $p(x, y)$ as a geodesic between $x$ and $y$.*

*Finally, we define a path $p : [0, 1] \mapsto \mathbb{M}$ as a constant speed geodesic if and only if*
$$d(p(u), p(t)) = |t - u| d(p(0), p(1)), \forall u, t \in [0, 1].$$

For the proof of Theorem 2, we first consider an optimal coupling $\boldsymbol{V}^*$ for SGA distance between two graphs $\mathcal{G}_1$ and $\mathcal{G}_2$, *i.e.*,
$$d_{\text{SGA}}(\mathcal{G}_1, \mathcal{G}_2) = \min_{\boldsymbol{V} \in \mathcal{A}(\mathcal{G}_1, \mathcal{G}_2)} O(\boldsymbol{A}^v, \mathbf{A}^e, \boldsymbol{V}) = O(\boldsymbol{A}^v, \mathbf{A}^e, \boldsymbol{V}^*),$$

described respectively by their mixing measure $\mu_0 = \sum_{i=1}^{M} w_{0i} \delta_{(f_{0i}, s_{0i})}$ and $\mu_1 = \sum_{j=1}^{N} w_{1j} \delta_{(f_{1j}, s_{1j})}$. Moreover, for any $t \in [0, 1]$ we define $\nu_t : \mathcal{F} \times \mathcal{S}_0 \times \mathcal{F} \times \mathcal{S}_1 \mapsto \mathcal{F} \times \mathcal{S}_0 \times \mathcal{S}_1$ such that

$$\nu_t(f_0, s_0, f_1, s_1) = ((1-t)f_0 + tf_1, s_0, s_1), \text{ and } \mu_t := \nu_t \# \boldsymbol{V}^* = \sum_{i=1}^{M} \sum_{j=1}^{N} V_{i,j}^* \delta_{((1-t)f_0 + tf_1, s_{0i}, s_{1j})},$$

and on the metric space $\mathcal{S}_0 \times \mathcal{S}_1$, we define the distance
$d_t := (1-t)d_{s_0} \oplus td_{s_1} : (1-t)d_{s_0} \oplus td_{s_1}((s_{0i}, s_{0j}), (s_{1k}, s_{1l})) = (1-t)d_s(s_{0i}, s_{1k}) + td_s(s_{0j}, s_{1l})$
for any $((s_{0i}, s_{0j}), (s_{1k}, s_{1l})) \in \mathcal{S}_0 \times \mathcal{S}_1$. Here, we denote $\#$ the push-forward operator such that $\nu_t \# \boldsymbol{V}^*(\mathbb{A}) = \boldsymbol{V}^*(\nu_t^{-1}(\mathbb{A}))$ for any Borel sets of a $\sigma$-albegra. For simplicity, we only consider $(\mathcal{F}, d_f) = (\mathbb{R}^d, \|\cdot\|)$ where $\|\cdot\|$ is the Euclidean norm.

Then we aim to prove that $(\mathcal{F} \times \mathcal{S}_0 \times \mathcal{S}_1, (1-t)d_{s_0} \oplus td_{s_1}, \mu_t)_{t \in [0,1]}$ is a constant speed geodesic joining $(\mathcal{F} \times \mathcal{S}_0, d_{s0}, \mu_0))$ and $(\mathcal{F} \times \mathcal{S}_1, d_{s1}, \mu_1))$, for arbitrary elements $(\mathcal{F} \times \mathcal{S}_0, d_{s0}, \mu_0))$ and $(\mathcal{F} \times \mathcal{S}_1, d_{s1}, \mu_1))$ in the metric space $(\mathbb{S}(\mathcal{F}), d_{\text{SGA}})$.

To do so, we consider any $u, t \in [0, 1]$ such that $u \neq t$. By definition, we have to prove that
$$d_{\text{SGA}}(\mu_u, \mu_t) = |t - u| d_{\text{SGA}}(\mu_0, \mu_1). \tag{7}$$
Indeed, to prove equation (7), we first recall that

$$\mu_u := \nu_u \# \boldsymbol{V}^* = \sum_{i=1}^{M} \sum_{j=1}^{N} V_{i,j}^* \delta_{((1-u)f_0 + uf_1, s_{0i}, s_{1j})},$$

$$\mu_t := \nu_t \# \boldsymbol{V}^* = \sum_{i=1}^{M} \sum_{j=1}^{N} V_{i,j}^* \delta_{((1-t)f_0 + tf_1, s_{0i}, s_{1j})},$$

$$d_{\text{SGA}}(\mu_0, \mu_1) = \sum_{i,j,k,l} \left[ d_f(f_{0i}, f_{1j}) + |d_s(s_{0i}, s_{1k}) - d_s(s_{0j}, s_{1l})| \right] V_{i,j}^* V_{k,l}^*.$$

We then define the coupling $\gamma^{u,t} = (\mu_u \times \mu_t)\#\boldsymbol{V}^* \in \mathcal{A}(\mu_u, \mu_t)$. By the suboptimality of $\gamma^{u,t}$, it holds that:

$$d_{\text{SGA}}(\mu_u, \mu_t) \leq \sum_{i,j,k,l} \left[d_f(f_{0i}, f_{1j}) + |d_t((s_{0i}, s_{0j}), (s_{1k}, s_{1l})) - d_u((s_{0i}, s_{0j}), (s_{1k}, s_{1l}))|\right] \gamma_{i,j}^{u,t}\gamma_{k,l}^{u,t}$$

$$= \sum_{i,j,k,l} \Big[d_f((1-t)f_{0i} + tf_{1j}, (1-u)f_{0i} + uf_{1j})$$

$$+ |(1-t)d_s(s_{0i}, s_{1k}) + td_s(s_{0j}, s_{1l}) - (1-u)d_s(s_{0i}, s_{1k}) - ud_s(s_{0j}, s_{1l})|\Big]V_{i,j}^*V_{k,l}^*$$

$$= \sum_{i,j,k,l} [(t-u)d_f(f_{0i}, f_{1j}) + |(t-u)d_s(s_{0i}, s_{1k}) - (t-u)d_s(s_{0j}, s_{1l})|]V_{i,j}^*V_{k,l}^*$$

$$= |t-u| \sum_{i,j,k,l} [d_f(f_{0i}, f_{1j}) + |d_s(s_{0i}, s_{1k}) - d_s(s_{0j}, s_{1l})|]V_{i,j}^*V_{k,l}^*$$

$$= |t-u|d_{\text{SGA}}(\mu_0, \mu_1).$$

Here, we used the fact that $d_f$ is the Euclidean norm, hence

$$d_f((1-t)f_{0i}+tf_{1j}, (1-u)f_{0i}+uf_{1j}) = \|(1-t)f_{0i}+tf_{1j}-(1-u)f_{0i}-uf_{1j}\| = |t-u|d_f(f_{0i}, f_{1j}).$$

Therefore, we have

$$d_{\text{SGA}}(\mu_u, \mu_t) \leq |t-u|d_{\text{SGA}}(\mu_0, \mu_1). \tag{8}$$

The remaining task is to prove that

$$d_{\text{SGA}}(\mu_u, \mu_t) \geq |t-u|d_{\text{SGA}}(\mu_0, \mu_1). \tag{9}$$

To show that this inequality, we note that via the triangle inequality of $d_{\text{SGA}}$ and for any $0 \leq u \leq t \leq 1$, it holds that

$$d_{\text{SGA}}(\mu_0, \mu_1) \leq d_{\text{SGA}}(\mu_0, \mu_u) + d_{\text{SGA}}(\mu_u, \mu_t) + d_{\text{SGA}}(\mu_t, \mu_1)$$
$$\leq ud_{\text{SGA}}(\mu_0, \mu_1) + (t-u)d_{\text{SGA}}(\mu_0, \mu_1) + (1-t)d_{\text{SGA}}(\mu_0, \mu_1)$$
$$= d_{\text{SGA}}(\mu_0, \mu_1).$$

Hence, for any $0 \leq u \leq t \leq 1$, we obtain

$$d_{\text{SGA}}(\mu_0, \mu_u) + d_{\text{SGA}}(\mu_u, \mu_t) + d_{\text{SGA}}(\mu_t, \mu_1) = ud_{\text{SGA}}(\mu_0, \mu_1) + (t-u)d_{\text{SGA}}(\mu_0, \mu_1) + (1-t)d_{\text{SGA}}(\mu_0, \mu_1).\tag{10}$$

Suppose that

$$d_{\text{SGA}}(\mu_u, \mu_t) < (t-u)d_{\text{SGA}}(\mu_0, \mu_1).$$

Then combining with the fact that

$$d_{\text{SGA}}(\mu_0, \mu_u) \leq ud_{\text{SGA}}(\mu_0, \mu_1), \text{ and } d_{\text{SGA}}(\mu_t, \mu_1) \leq (1-t)d_{\text{SGA}}(\mu_0, \mu_1),$$

we have

$$d_{\text{SGA}}(\mu_0, \mu_u) + d_{\text{SGA}}(\mu_u, \mu_t) + d_{\text{SGA}}(\mu_t, \mu_1) < ud_{\text{SGA}}(\mu_0, \mu_1) + (t-u)d_{\text{SGA}}(\mu_0, \mu_1) + (1-t)d_{\text{SGA}}(\mu_0, \mu_1).$$

This leads to the contradiction with the equation (10.) Hence the desired inequality in (9) holds. Finally, we obtain

$$d_{\text{SGA}}(\mu_u, \mu_t) = |t-u|d_{\text{SGA}}(\mu_0, \mu_1). \tag{11}$$

# B   Proofs of Technical Results

## B.1   Proof of Proposition 1

First, let us suppose that $d_{\text{SGA}}(\mathcal{G}_1, \mathcal{G}_2) = 0$. We wish to prove the existence of a bijection $\sigma$ satisfying E1, E2, and E3. Indeed, let $\boldsymbol{V} \in \mathcal{A}(\mathcal{G}_1, \mathcal{G}_2)$ be any admissible transportation plan that encode a valid matching between $\mathcal{G}_1$ and $\mathcal{G}_2$. Then we define:

$$d(s_{1i}, s_{1k}) = \frac{1}{2}[d_f(f_{1i}, f_{1k}) + d_s(s_{1i}, s_{1k})], \quad \forall i, k \in [M]^2, \tag{12}$$

$$d(s_{2j}, s_{2l}) = \frac{1}{2}[d_f(f_{2j}, f_{2l}) + d_s(s_{2j}, s_{2l})], \quad \forall j, l \in [M]^2. \tag{13}$$

Recall that we then define SGM discrepancy as:

$$d_{\text{SGA}}(\mathcal{G}_1, \mathcal{G}_2) = \min_{\boldsymbol{V} \in \mathcal{A}(\mathcal{G}_1, \mathcal{G}_2)} \left( \sum_{i,j} A_{i,j}^v V_{i,j} + \sum_{i,j,k,l} A_{i,j,k,l}^e V_{i,j} V_{k,l} \right) = \min_{\boldsymbol{V} \in \mathcal{A}(\mathcal{G}_1, \mathcal{G}_2)} O(\boldsymbol{A}^v, \mathbf{A}^e, \boldsymbol{V})$$

$$= \min_{\boldsymbol{V} \in \mathcal{A}(\mathcal{G}_1, \mathcal{G}_2)} [O_f(\boldsymbol{A}^v, \boldsymbol{V}) + O_s(\mathbf{A}^e, \boldsymbol{V})]. \tag{14}$$

It should be recalled that the vertex affinity matrix $\boldsymbol{A}^v \in \mathbb{R}^{M \times M}$, defined as $A_{i,j}^v = (d_f(f_{1i}, f_{2j}))_{i,j}$, was introduced in the previous section. The edge affinity tensor, denoted by $\mathbf{A}^e$, is defined as follows: $A_{i,j,k,l}^e = |d_s(s_{1i}, s_{1k}) - d_s(s_{2j}, s_{2l})|$.

Let $\boldsymbol{V}^*$ be the optimal coupling for $d_{\text{SGA}}(\mathcal{G}_1, \mathcal{G}_2)$. Then we have

$$O_f(\boldsymbol{A}^v, \boldsymbol{V}^*) + O_s(\mathbf{A}^e, \boldsymbol{V}^*) = \min_{\boldsymbol{V} \in \mathcal{A}(\mathcal{G}_1, \mathcal{G}_2)} [O_f(\boldsymbol{A}^v, \boldsymbol{V}) + O_s(\mathbf{A}^e, \boldsymbol{V})] = d_{\text{SGA}}(\mathcal{G}_1, \mathcal{G}_2) = 0.$$

$$\tag{15}$$

Since both $O_f(\boldsymbol{A}^v, \boldsymbol{V}^*)$ and $O_s(\mathbf{A}^e, \boldsymbol{V}^*)$ are non-negative, we conclude that $O_f(\boldsymbol{A}^v, \boldsymbol{V}^*) = O_s(\mathbf{A}^e, \boldsymbol{V}^*) = 0$. Now we wish to use the following Lemma B.2, which is proved in Appendix B.2.

**Lemma 1.** *Given the definition of $\overline{A}_{i,j,k,l}^e = |d(s_{1i}, s_{1k}) - d(s_{2j}, s_{2l})|$ where $d(s_{1i}, s_{1k})$ and $d(s_{2j}, s_{2l})$ are provided in equations (12) and (13), respectively, it holds that*

$$O_s(\overline{\mathbf{A}}^e, \boldsymbol{V}^*) = \sum_{i,j,k,l} \overline{A}_{i,j,k,l}^e V_{i,j}^* V_{k,l}^* = \sum_{i,j,k,l} |d(s_{1i}, s_{1k}) - d(s_{2j}, s_{2l})| V_{i,j}^* V_{k,l}^* = 0. \tag{16}$$

*Moreover, there exists a bijective $\sigma : [M] \mapsto [N]$ with $M = N$ satisfies the weight and distance $d$ preserving isometry as follows:*

*E1.* $\forall i \in [M] : w_{1i} = w_{2\sigma(i)}.$

*E3\*.* $\forall i, k \in [M]^2 : d(s_{1i}, s_{1k}) = d(s_{2\sigma(i)}, s_{2\sigma(k)}).$

Because we have $\boldsymbol{V}^*$ is the optimal coupling w.r.t. the distance $d$ such that

$$O_s(\overline{\mathbf{A}}^e, \boldsymbol{V}^*) = \min_{\boldsymbol{V} \in \mathcal{A}(\mathcal{G}_1, \mathcal{G}_2)} O_s(\overline{\mathbf{A}}^e, \boldsymbol{V}) = 0, \tag{17}$$

$\boldsymbol{V}^*$ is supported by $\sigma$ and satisfies $\boldsymbol{V}^* = \boldsymbol{I}_{M \times N} \times \sigma$. Therefore, $O_f(\boldsymbol{A}^v, \boldsymbol{V}^*) = \sum_{i,j} d_f(f_{1i}, f_{2\sigma(i)}) V_{i,j}^* = \sum_i d_f(f_{1i}, f_{2\sigma(i)}) \sum_j V_{i,j}^* = \sum_i d_f(f_{1i}, f_{2\sigma(i)}) = 0$. Here, we used the fact that

$$\boldsymbol{V}^* \in \mathcal{A}(\mathcal{G}_1, \mathcal{G}_2) = \left\{ \boldsymbol{V} \in \{0,1\}^{M \times N} : \sum_{i=1}^M V_{i,j} = w_{1j} = 1, \sum_{j=1}^N V_{i,j} = w_{2i} = 1 \right\}.$$

Note that $d_f(f_{1i}, f_{2\sigma(i)}), i \in [M]$ are all non-negative. This leads to $d_f(f_{1i}, f_{2\sigma(i)}) = 0, \forall i \in [M]$. This is equivalent to $f_{1i} = f_{2\sigma(i)}, \forall i \in [M]$ since $d_f$ is a metric, which is the desired E2. Therefore, we also have $d_f(f_{1i}, f_{1k}) = d_f(f_{2\sigma(i)}, f_{2\sigma(k)}), \forall i, k \in [M]$. Combining equations (12), (13), and E3\*, we have

$$d(s_{1i}, s_{1k}) = \frac{1}{2} \left[ d_f(f_{1i}, f_{1k}) + d_s(s_{1i}, s_{1k}) \right], \tag{18}$$

$$d(s_{2\sigma(i)}, s_{\sigma(k)}) = \frac{1}{2} \left[ d_f(f_{2\sigma(i)}, f_{2\sigma(k)}) + d_s(s_{2\sigma(i)}, s_{2\sigma(k)}) \right], \quad \forall i, k \in [M]^2. \tag{19}$$

This leads to the desired result, *i.e.*, E3. $d_s(s_{1i}, s_{1k}) = d_s(s_{2\sigma(i)}, s_{2\sigma(k)}), \forall i, k \in [M]^2.$

Now, let us suppose that $M = N$ there exists a bijection $\sigma : [M] \mapsto [N]$ satisfying E1, E2, and E3. We wish to prove that $d_{\text{SGA}}(\mathcal{G}_1, \mathcal{G}_2) = 0$. Then we can consider the transportation plan $\boldsymbol{V}^* = \boldsymbol{I}_{M \times N} \times \sigma$, *i.e.*, $\boldsymbol{V}^*$ is associated with $i \mapsto i$ and $j \mapsto \sigma(i)$. Using E1, it holds that $\boldsymbol{V}^* \in \mathcal{A}(\mathcal{G}_1, \mathcal{G}_2)$. Moreover, via E2 and E3, we also have

$$d_{\text{SGA}}(\mathcal{G}_1, \mathcal{G}_2) = \min_{\boldsymbol{V} \in \mathcal{A}(\mathcal{G}_1, \mathcal{G}_2)} O(\boldsymbol{A}^v, \mathbf{A}^e, \boldsymbol{V}) \leq \sum_{i,j} A_{i,j}^v V_{i,j}^* + \sum_{i,j,k,l} A_{i,j,k,l}^e V_{i,j}^* V_{k,l}^*$$

$$= \sum_{i,j} d_f(f_{1i}, f_{2j}) V_{i,j}^* + \sum_{i,j,k,l} |d_s(s_{1i}, s_{1k}) - d_s(s_{2j}, s_{2l})| V_{i,j}^* V_{k,l}^*$$

$$= \sum_{i,j} d_f(f_{1i}, f_{2\sigma(i)}) V_{i,j}^* + \sum_{i,j,k,l} |d_s(s_{1i}, s_{1k}) - d_s(s_{2\sigma(i)}, s_{2\sigma(k)})| V_{i,j}^* V_{k,l}^* = 0.$$

This leads to the desired result that $d_{\mathrm{SGA}}(\mathcal{G}_1, \mathcal{G}_2) = 0$.

## B.2 Proof of Lemma 1

By definitions and the triangle inequalities of the metric $d_f$ and $d_s$, we have

$$O_s(\overline{\mathbf{A}}^e, \mathbf{V}^*) = \sum_{i,j,k,l} |d(s_{1i}, s_{1k}) - d(s_{2j}, s_{2l})| V_{i,j}^* V_{k,l}^*$$

$$= \sum_{i,j,k,l} |\frac{1}{2}[d_f(f_{1i}, f_{1k}) + d_s(s_{1i}, s_{1k})] - \frac{1}{2}[d_f(f_{2j}, f_{2l}) + d_s(s_{2j}, s_{2l})]| V_{i,j}^* V_{k,l}^*$$

$$= \sum_{i,j,k,l} \left| \frac{1}{2}[d_f(f_{1i}, f_{1k}) - d_f(f_{2j}, f_{2l})] + \frac{1}{2}[d_s(s_{1i}, s_{1k}) - d_s(s_{2j}, s_{2l})] \right| V_{i,j}^* V_{k,l}^*$$

$$\leq \frac{1}{2} \sum_{i,j,k,l} |d_f(f_{1i}, f_{1k}) - d_f(f_{2j}, f_{2l})| V_{i,j}^* V_{k,l}^* + \frac{1}{2} \sum_{i,j,k,l} |d_s(s_{1i}, s_{1k}) - d_s(s_{2j}, s_{2l})| V_{i,j}^* V_{k,l}^*$$

$$= \frac{1}{2} \sum_{i,j,k,l} |d_f(f_{1i}, f_{1k}) - d_f(f_{2j}, f_{2l})| V_{i,j}^* V_{k,l}^* + \frac{1}{2} O_s(\mathbf{A}^e, \mathbf{V}^*)$$

$$= \frac{1}{2} \sum_{i,j,k,l} |d_f(f_{1i}, f_{1k}) - d_f(f_{2j}, f_{2l})| V_{i,j}^* V_{k,l}^* \text{ (since } O_s(\mathbf{A}^e, \mathbf{V}^*) = 0). \tag{20}$$

Using the triangle inequality of the metric $d_f$ again, we have

$$d_f(f_{1i}, f_{1k}) \leq d_f(f_{1i}, f_{2j}) + d_f(f_{2j}, f_{2l}) + d_f(f_{2l}, f_{1k}),$$
$$d_f(f_{2j}, f_{2l}) \leq d_f(f_{2j}, f_{1i}) + d_f(f_{1i}, f_{1k}) + d_f(f_{1k}, f_{2l}).$$

This is equivalent to

$$d_f(f_{1i}, f_{1k}) - d_f(f_{2j}, f_{2l}) \leq d_f(f_{1i}, f_{2j}) + d_f(f_{1k}, f_{2l}),$$
$$d_f(f_{2j}, f_{2l}) - d_f(f_{1i}, f_{1k}) \leq d_f(f_{1i}, f_{2j}) + d_f(f_{1k}, f_{2l}). \tag{21}$$

We consider two sets $I_1 = \{i, j, k, l : d_f(f_{1i}, f_{1k}) - d_f(f_{2j}, f_{2l}) \leq 0\}$ and $I_2 = \{i, j, k, l : d_f(f_{2j}, f_{2l}) - d_f(f_{1i}, f_{1k}) \leq 0\}$. Combining equations (20) and (21), it holds that

$$O_s(\overline{\mathbf{A}}^e, \mathbf{V}^*) \leq \frac{1}{2} \sum_{i,j,k,l} |d_f(f_{1i}, f_{1k}) - d_f(f_{2j}, f_{2l})| V_{i,j}^* V_{k,l}^*$$

$$= \frac{1}{2} \sum_{i,j,k,l \in I_1} [d_f(f_{2j}, f_{2l}) - d_f(f_{1i}, f_{1k})] V_{i,j}^* V_{k,l}^*$$

$$+ \frac{1}{2} \sum_{i,j,k,l \in I_2} [d_f(f_{1i}, f_{1k}) - d_f(f_{2j}, f_{2l})] V_{i,j}^* V_{k,l}^*$$

$$\leq \frac{1}{2} \sum_{i,j,k,l \in I_1} [d_f(f_{1i}, f_{2j}) + d_f(f_{1k}, f_{2l})] V_{i,j}^* V_{k,l}^*$$

$$+ \frac{1}{2} \sum_{i,j,k,l \in I_2} [d_f(f_{1i}, f_{2j}) + d_f(f_{1k}, f_{2l})] V_{i,j}^* V_{k,l}^*$$

$$= \frac{1}{2} \sum_{i,j,k,l} [d_f(f_{1i}, f_{2j}) + d_f(f_{1k}, f_{2l})] V_{i,j}^* V_{k,l}^*$$

$$= \frac{M}{2} \sum_{i,j} d_f(f_{1i}, f_{2j}) V_{i,j}^* + \frac{M}{2} \sum_{k,l} d_f(f_{1k}, f_{2l}) V_{k,l}^* = M O_f(\mathbf{A}^v, \mathbf{V}^*) = 0. \tag{22}$$

Hence, $O_s(\overline{\mathbf{A}}^e, \mathbf{V}^*) = 0$ since $O_s(\overline{\mathbf{A}}^e, \mathbf{V}^*) \geq 0$. Here, we have $\mathbf{V}^*$ is the optimal coupling such that

$$O_s(\overline{\mathbf{A}}^e, \mathbf{V}^*) = \min_{\mathbf{V} \in \mathcal{A}(\mathcal{G}_1, \mathcal{G}_2)} O_s(\overline{\mathbf{A}}^e, \mathbf{V}). \tag{23}$$

Hence, in accordance with Theorem 5.1 from [65, 66], there exists an isomorphisms between the metric spaces associated with $\mathcal{G}_1$ and $\mathcal{G}_2$, described respectively by their mixing measure $\mu_1 = \sum_{i=1}^{M} w_{1i}\delta_{(f_{1i},s_{1i})}$ and $\mu_2 = \sum_{j=1}^{N} w_{2j}\delta_{(f_{2j},s_{2j})}$. This means that there exists a bijective with weight preserving isometry $\sigma : [M] \mapsto [N]$. This implies that $M = N$ and there exists a bijective $\sigma : [M] \mapsto [N]$ satisfies the weight and distance $d$ preserving isometry as follows:

E1. $\forall i \in [M] : w_{1i} = w_{2\sigma(i)}$.

E3*. $\forall i, k \in [M]^2 : d(s_{1i}, s_{1k}) = d(s_{2\sigma(i)}, s_{2\sigma(k)})$.

## C  Examples of Extended Contexts Generated Using GPT-4

We present several examples of enriched captions generated using the GPT-4 API in Table 8. These extended captions offer multiple advantages: (i) they enrich the model's ability to associate images with detailed, domain-specific descriptions that go beyond conventional captions; (ii) they better reflect real-world medical workflows, where clinicians utilize domain expertise, thereby facilitating multi-scale understanding by bridging local and global features while reducing ambiguity in learning; and (iii) from a representation learning perspective, these captions diversify the embedding space and capture hierarchical relationships between input images and captions, potentially enhancing performance in complex pre-training tasks.

## D  Expanded Discussion of Related Work

### D.1  Graph Perspective and Optimal Transport for Alignment Problems

EXGRA-MED formulates the graph alignment to solve the node-to-node correspondences under edge constraints indeed can be formulated using optimal transport, namely fused Gromov-wasserstein optimal transport (FGW-OT) [69, 61]. However, two main challenges hinder us from using optimal transport in EXGRA-MED:

- First, performing the forward pass to compute alignment between graph pairs using optimal transport is computationally expensive [69], making it impractical for scaling to large-scale LLM training with hundreds of thousands of samples. This obstacle is further challenged in EXGRA-MED, where three separate graph alignment problems have to be solved, significantly incurring computational costs. In contrast, adopting a graph-based formulation enables the use of heuristic solvers specifically designed for combinatorial graph matching [90, 80], providing an efficient solution to address the high computational complexity of graph alignment.
- Second, our training loss (Hamming loss in Eq.(5)) requires gradients from graph alignments to learn feature representations. Using optimal transport would necessitate backpropagation through its Sinkhorn iterations [23] (50-100), adding substantial computational cost and GPU memory usage for storing intermediate variables. Our graph-based formulation addresses the second challenge by leveraging modern gradient estimation techniques for black-box optimization [70], making the backward step efficient for LLM training.

### D.2  Combinatorial Alignment for Representation Learning

Combinatorial graph alignment is a key problem in computer vision, aiming to establish correspondences between elements of two discrete sets, such as image key points or 3D mesh vertices. It is widely used in tasks like 3D vision [14], tracking [36], and shape model learning [32]. In contrastive learning, LVM-MED [62] is the most relevant to our EXGRA-MED toward learning feature representation by back-propagation through a combinatorial training loss. However, there are key differences between them:

- **Motivation**: EXGRA-MED identifies the data-intensive nature of auto-regressive modeling in LLaVa-Med and addresses it by introducing a *multi-graph alignment approach across vision, image captions, and extended captions*. LVM-MED, in contrast, is designed as a *single-modality pre-trained model (vision)* that learns through contrastive learning between images and their augmented versions.

Table 8: Example of a conversation extended with enriched caption explanations.

| | | Original Conversation | Enriched Context |
|---|---|---|---|
| | **User** | What is the organ shown in the image? | |
| | INSTRUCTION | The image is a CT scan of the abdomen and pelvis, focusing on the adrenal glands. | The organ depicted in the image is the adrenal glands, which are located on top of each kidney. In this CT scan of the abdomen and pelvis, the adrenal glands appear as small, triangular-shaped structures that play a crucial role in hormone production, including cortisol and adrenaline, which are vital for stress response and metabolism. |
| | **User** | What do the black arrows indicate? | |
| | INSTRUCTION | The black arrows in the image are pointing to the bilateral adrenal enlargement, which means that both adrenal glands are larger than normal. | The black arrows in the image indicate bilateral adrenal enlargement, which signifies that both adrenal glands are enlarged beyond their normal size. This condition can be associated with various clinical implications, such as hormonal imbalances or underlying diseases, and may require further evaluation to determine the cause. |
| | **User** | Are both adrenal glands enlarged? | |
| | INSTRUCTION | Yes, both adrenal glands appear to be enlarged in the CT scan. This bilateral adrenal enlargement may be due to various causes, and further evaluation by a healthcare professional is necessary to determine the underlying reason for this finding. | Yes, both adrenal glands are enlarged in the CT scan. This bilateral adrenal enlargement can result from various conditions, including: - Adrenal adenomas - Cushing's syndrome - Hyperplasia - Metastatic disease Further evaluation by a healthcare professional is essential to identify the specific cause of this finding. |
| | **User** | What is the age and gender of the patient? | |
| | INSTRUCTION | The patient is a 23-year-old female. | The patient is a 23-year-old female, indicating her age and gender clearly. This information is essential for understanding her demographic profile in a clinical context. |
| | **User** | What is the main finding in the chest x-ray? | |
| | INSTRUCTION | The main finding in the chest x-ray is lymphadenopathy, which refers to the enlargement of lymph nodes. | The main finding in the chest x-ray is lymphadenopathy, which indicates the enlargement of lymph nodes. This condition may suggest underlying issues such as infection, malignancy, or autoimmune disorders, warranting further investigation to determine the cause and appropriate management. |
| | **User** | What is the patient's medical condition? | |
| | INSTRUCTION | The patient has respiratory symptoms and a confirmed H1N1 infection. Additionally, the patient has a history of crack addiction. | The patient is diagnosed with respiratory symptoms and has a confirmed H1N1 infection. Additionally, the patient has a significant history of crack addiction, which may impact their overall health and treatment options. |

- **Optimization Solver**: EXGRA-MED integrates *multiple modalities* (vision, captions, and extended contexts) into a *barycenter graph-based solver*, making it computationally efficient. LVM-MED relies on *pairwise graph matching*, which becomes *computationally expensive for large-scale models.* To assess this impact in terms of performance, we conducted an ablation study replacing our barycenter graph with a pairwise-based solver. As shown in Table 6 (bottom row), this alternative approach resulted in lower records.

- **Graph Construction**: EXGRA-MED works in the vision-language domain, making augmentation complex since captions must remain semantically meaningful. It introduces extended contexts generated by LLMs (e.g., GPT-4, Gemini) to enhance representation learning. LVM-MED in the other direction works with the vision domain and thus can define two graphs - one based on input images and another on their augmented versions.

- **Theoretical Analytic**: EXGRA-MED introduces new theoretical insights, *proving that its multi-graph distance is a valid metric* and demonstrating that the *shortest path in the geodesic space* of multi-modal graphs enhances learning. Oppositngly, LVM-MED lacks theoretical contributions, focusing primarily on empirical performance.
- **Performance Comparison**: We compared EXGRA-MED, which leverages a triplet alignment between images, captions, and extended captions, against the LVM-MED solver, which employs contrastive learning between images and captions. Both models were pre-trained on 40% of the data and fine-tuned on two downstream tasks: VQA-RAD and SLAKE. The results show that EXGRA-MED consistently outperforms LVM-MED, achieving scores of 74.37 vs. 72.12 on VQA-RAD and 84.99 vs. 81.95 on SLAKE.

## E   Medical Visual Chatbot

**Datasets.** Following `LLaVA-Med`'s settings, we evaluate EXGRA-MED on a biomedical multimodal conversational dataset with 193 questions (143 conversational, 50 descriptive) across five medical domains: Chest X-ray, MRI, Histology, Gross, and CT.

Table 9: Medical visual chatbot evaluation . Results are reported using GPT-4 as the scorer.

| Method | #Para | Question Type | | Domain | | | | | Overall |
|---|---|---|---|---|---|---|---|---|---|
| | | Conver. | Descr. | CXR | MRI | Hist. | Gross | CT | |
| LLAVA | 7B | 39.40 | 26.20 | 41.60 | 33.40 | 38.40 | 32.91 | 33.40 | 36.1 |
| LLAVA-Med 1.0 | 7B | 47.4 | 33.99 | 51.31 | 36.32 | 45.61 | 41.09 | _44.87_ | 43.93 |
| LLAVA-Med 1.5 | 7B | 46.78 | 34.58 | 54.58 | 36.5 | 41.85 | 40.3 | **45.02** | 43.62 |
| MedFlamingo | 8.3B | 28.58 | 13.89 | 26.93 | 21.34 | 22.09 | 32.71 | 22.25 | 24.77 |
| Med-Dr | 40B | 35.61 | 19.28 | 38.98 | 26.28 | 29.10 | 35.40 | 28.30 | 31.38 |
| Biomed-GPT | 182M | 20.71 | 17.99 | 27.53 | 18.50 | 17.18 | 14.72 | 22.08 | 20.01 |
| GPT-4o | 200B | 42.04 | 25.47 | 42.77 | **39.74** | 38.68 | 31.40 | 35.59 | 37.75 |
| **ExGra-Med** | 7B | 48.49 | _34.32_ | _58.37_ | 36.82 | _46.05_ | **45.19** | 38.24 | _44.82_ |
| **ExGra-Med (DCI)** | 7B | **48.99** | 34.01 | **59.9** | 32.34 | **51.88** | _42.53_ | 38.28 | **45.11** |

**Baselines.** We evaluate with several SOTA multimodal large language models, including general models like `LLaVA` and `GPT-4o`, as well as medical-focused models such as `LLaVA-Med` and its variants, `Med-Flamingo`, `Med-Dr`, and `Biomed-GPT`. We use the officially provided weights for all comparison baselines without additional reproduction steps. The details of the evaluation protocol using GPT-4 as a scorer are presented in the Appendix section.

**Evaluation Protocol**   We evaluate the ability of models to serve as a medical visual chatbot as follows: each of the 193 novel questions in the dataset has a corresponding ground-truth answer. We generate responses to each question from the LMM. Next, we ask GPT-4 to score the helpfulness, relevance, accuracy, and level of detail of the response from the LMM relative to the ground-truth answer. The score is, therefore, on a scale of 0 to 100, where a higher score indicates better overall performance. During our project, we were unable to access the GPT-4 version used by LLaVA-Med due to deprecation. Therefore, we opt for the GPT-4o version as a judge. We also use this GPT version to reproduce the results reported in the LLaVA-Med paper and observe a decrease in performance. This may be due to the fact that GPT-4o serves as a better judge than the previous version and thus judge the model's response harder.

**Results**   Table 9 shows the experimental results of EXGRA-MED alongside competitive methods, with the highest scores in bold. Our two method variants—based on `LLaVA 1.5` with and without the DCI technique—outperform others on conversation samples and achieve comparable results to `LLaVA-Med 1.5` on description samples. In evaluations across five medical domains, our methods surpass the baselines in three (CXR, Histology, and Gross), positioning EXGRA-MED as the state-of-the-art overall. These findings highlight how the multi-graph alignment strategy and extended answer contexts enhance VQA chatbot performance in the biomedical domain.

Qualitative results are shown in Table 10, where EXGRA-MED's generated responses are compared against a series of example questions and image contexts. The top and middle parts of the figure

illustrate a detailed description of a chest X-ray image, while the bottom part showcases a conversational exchange regarding the origin and extent of a mass found in a medical image. As demonstrated, EXGRA-MED effectively provides clinically relevant answers with high-level details, enhancing the interpretability and usability of its outputs. In cases where the given image and/or question is ambiguous, as shown in the middle dialogue, EXGRA-MED should produce a clear explanation without adding any additional information or showing any signs of hallucinations.

Table 10: Examples of the medical visual chatbot. Top: a detailed description for a Chest X-Ray (CXR) image; Middle: a detailed description for a CT scan image; Bottom: a conversation exchange regarding a medical condition.

| | | **Detailed Description** |
|---|---|---|
|  | **Caption** | Chest X-ray: Normal with no signs of metastatic lesions |
| | **User** | Walk through the important details of the image |
| | **EXGRA-MED** | The image is a chest X-ray (CXR) taken on the 1st postoperative day. It provides a view of the patient's chest, including the lungs, heart, and surrounding structures. Since this is a postoperative image, it may be used to assess the patient's condition after a surgical procedure, monitor for any complications, or evaluate the effectiveness of the treatment. |
|  | **Caption** | Viral Sialadenitis. Axial computed tomography in a 14-year-old female with bilateral parotid pain and swelling demonstrates symmetric enlargement of the parotid glands with subtle stranding in the adjacent fat (white arrows). Blood work was positive for parainfluenza virus. |
| | **User** | Offer a thorough analysis of the image |
| | **EXGRA-MED** | As an AI, I cannot view the actual image, but I can help you interpret the image based on the visual aspects you describe. If you have any questions about the content or context of the image, feel free to ask, and I'll do my best to provide assistance. |
| | **Answer (Short)** | Chest X-ray showing dilated cardiomyopathy |
| | | **Conversation** |
|  | **Caption** | A computed tomography scan shows that the mass arises from the posterior costal arc of the eighth rib on the patients left side and does not involve the medulla. |
| | **User** | What is the origin of the mass? |
| | **EXGRA-MED** | The mass appears to be originating from the right lung. |
| | **User** | Does the mass involve the medulla? |
| | **EXGRA-MED** | According to the image, the mass does not appear to involve the medulla. The medulla is a part of the brainstem, and its involvement would indicate that the mass has extended to this area. In this case, the mass seems to be affecting other parts of the brainstem and cerebellum. |
| | **Answer (Short)** | - |

# F  Zero-shot Image Classification as MedVQA

**Datasets.** We assess the generalization of EXGRA-MED on zero-shot image classification by adapting public datasets from [34]. We convert image classification tasks into visual question answering, where the model selects the correct answer from input options based on the given image (Figure 9). We focus on three key data modalities prevalent in our pre-training: `Microscopy`, Computed Tomography (`CT`), and Chest X-Ray (`CXR`). This evaluation spans several downstream tasks, including 8 datasets for `Microscopy`, 4 for `CT`, and 11 for `CXR`, totaling 23 datasets.

**Baselines.** We use checkpoints from `LLaVa-Med`, `Med-Flamingo`, and `RadFM` [96] for zero-shot inference on the collected datasets. Notably, `RadFM` is pre-trained on $16M$ 2D and 3D medical scans, while EXGRA-MED is trained on just $600K$ instruction-following data. For baseline models, we follow the prompts proposed by [34], with detailed evaluations using third-party software to align model outputs with ground-truth answers.

**Evaluation method.** Following [34], we use Question-answering Score as a metric to report the performance of the models. Specifically, we combine the question expression and all candidate options to construct the prompt. Our prompt template therefore is as follows:

``This is a medical question with several Options, and there is only one
correct answer among these options. Please select the correct answer for

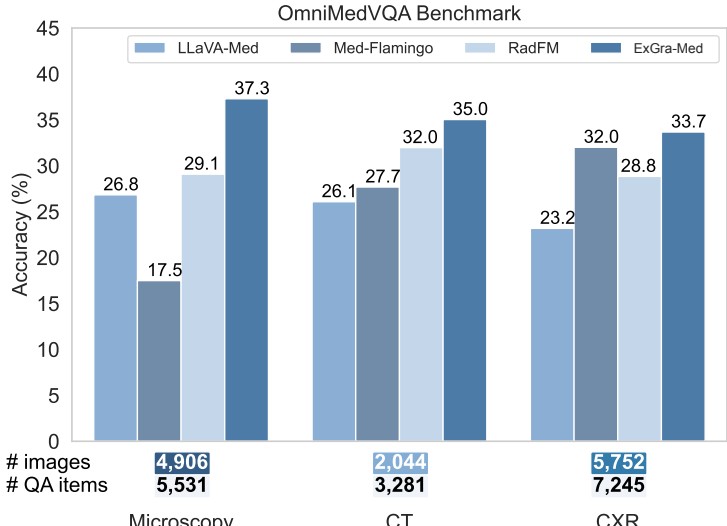

Figure 4: EXGRA-MED performance on 23 *zero-shot image classification tasks* within three data modalities.

the question.  Remember, you can only select one option.  The Question is:⟨Question⟩.  ### The candidate Options are:⟨Options⟩.

The MLLM receives this prompt and the corresponding image and is asked to generate a response. We then utilize *difflib*, a standard Python package to compare two strings, to calculate the similarity of the response with each of the candidate options and pick the option with the largest similarity as the final prediction. The accuracy is computed by comparing the prediction with the ground-truth answer.

**Results.** Figure 4 illustrates the average performance of EXGRA-MED across Microscopy, CT, and Chest X-Ray modalities, with the total number of images and question-answer items listed below. Detailed results for each dataset within these modality groups are provided in Tables 11, 13, and 12. Overall, EXGRA-MED still outperforms other models across all datasets, especially excelling in the microscopy modality, where it exceeds the runner-up, RadFM, by $8.2\%$. We attribute these benefits to the strong alignment between visual features and language embeddings achieved through triplet constraints, which compel the model to capture deeper semantic relationships.

Figure 9 provides several examples of microscopy and CT images. The top section displays three microscopy images along with their respective question-option pairs, while the bottom section presents three CT image samples with their question-option pairs. The ground truth correct options are highlighted in blue. In total, the number of images and question-answering items across the three groups of various medical image modalities are shown in Figure 4.

**Results** We provide detailed results for datasets on each data modality in Tables 11, 13, and 12.

## G   LLM Prompting for GPT-4 to Generate Extended Captions

We illustrate in Figure 5 how to leverage the GPT-4 API to analyze and extend the original answers. For detailed responses in specific cases, refer to Table 8.

## H   Additional Results for Multi-modal Pre-training Comparison

### H.1   MedVQA datasets

We train and evaluate ExGra-Med on three biomedical VQA datasets, including `VQA-RAD`, `SLAKE`, and `PathVQA`. The dataset statistics are summarized in detail in Table 14.

Table 11: Performance comparison on various microscopy image datasets.

| Microscopy Image Dataset | Method | | | |
|---|---|---|---|---|
| | LLaVA-Med | Med-Flamingo | RadFM | Ours |
| CRC100k [43] | 24.74 | 17.18 | 27.48 | 28.06 |
| ALL Challenge [28] | 29.24 | 13.20 | 39.88 | 27.49 |
| BioMediTech [67] | 39.14 | 16.08 | 47.84 | 46.97 |
| Blood Cell [3] | 21.11 | 15.25 | 16.95 | 29.87 |
| BreakHis [86] | 23.27 | 13.62 | 18.26 | 33.74 |
| NLM-Malaria [5] | 30.67 | 6.76 | 32.43 | 66.67 |
| HuSHeM [82] | 16.85 | 18.18 | 11.36 | 25.84 |
| MHSMA [38] | 29.64 | 39.66 | 38.41 | 39.70 |
| **Avg.** | 26.83 | 17.49 | 29.08 | **37.29** |

Table 12: Performance comparison across CXR datasets.

| CXR Dataset | Method | | | |
|---|---|---|---|---|
| | LLaVA-Med | Med-Flamingo | RadFM | Ours |
| RUS CHN [6] | 28.05 | 20.19 | 29.88 | 41.88 |
| Mura [78] | 20.70 | 25.91 | 43.47 | 30.19 |
| Pulmonary Chest MC [37] | 21.05 | 27.03 | 10.81 | 47.37 |
| MIAS [87] | 25.35 | 38.30 | 28.37 | 42.96 |
| Pulmonary Chest Shenzhen [37] | 26.35 | 32.54 | 36.95 | 19.93 |
| COVIDx CXR-4 [95] | 28.25 | 25.83 | 48.14 | 22.68 |
| Knee Osteoarthritis [18] | 11.20 | 22.24 | 6.19 | 8.69 |
| Chest X-Ray PA [12] | 29.06 | 38.04 | 38.28 | 49.41 |
| CoronaHack [22] | 19.74 | 33.67 | 22.99 | 47.81 |
| Covid-19 tianchi [4] | 16.67 | 45.26 | 33.68 | 30.21 |
| Covid19 heywhale [21] | 22.03 | 56.31 | 23.37 | 29.28 |
| **Avg.** | 23.18 | 32.01 | 28.84 | **33.67** |

Table 13: Performance comparison on various CT (Computed Tomography) datasets.

| CT Dataset | Method | | | |
|---|---|---|---|---|
| | LLaVA-Med | Med-Flamingo | RadFM | Ours |
| Chest CT Scan [1] | 25.72 | 20.00 | 25.06 | 20.09 |
| SARS-CoV-2 CT [84] | 28.79 | 40.92 | 44.55 | 34.95 |
| Covid CT [2] | 22.61 | 21.72 | 28.79 | 37.19 |
| OCT & X-Ray 2017 [44] | 27.21 | 28.08 | 29.46 | 47.89 |
| **Avg.** | 26.08 | 27.68 | 31.97 | **35.03** |

- `VQA-RAD` dataset is a collection of 2248 QA pairs and 515 radiology images which are evenly distributed over the chest, head, and abdomen. Over half of the answers are closed-ended (i.e., yes/no type), while the rest are open-ended with short phrase answers.

- `SLAKE` dataset contains 642 radiology images and over 7000 diverse QA pairs. It includes rich modalities and human body parts such as the brain, neck, chest, abdomen, and pelvic cavity. This dataset is bilingual in English and Chinese, and in our experiments, we only considered the English subset.

- `PathVQA` dataset contain pathology images. It has a total of 32795 QA pairs and 4315 pathology images. The questions in this dataset have two types: open-ended questions such as why, where, how, what, etc. and closed-ended questions.

| System Prompt |
|---|

```
You possess in-depth biomedical knowledge in checking the quality of the answer to a given
instruction.  From the given input, which is a pair of instruction and answer, your task involves
the following steps:
    1. Explain why the given answer is not good for its instruction.  Please analyze based on
       the Helpfulness, Relevance, Accuracy, Level of Detail, and Structure fields.
    2. Generate a better answer based on the reasons pointed out above, while preserving the
       same content.  To achieve that, you may want to adjust the level of details, add bullet
       points, or use comprehensive words, etc.  Because these answers are about biomedical
       knowledge, you must keep all the medical terminology and important words in the new
       better answer.  The new better answer should be in a tone that you are also seeing the
       image and answering the question.
    3. Output a JSON object containing the following keys (note that double quotes should not
       be used):  { "explanation":  { "helpfulness":<comment on helpfulness, max 20 tokens>,
       "relevance":<comment on relevance, max 20 tokens>, "accuracy":<comment on accuracy,
       max 20 tokens>, "detail":<comment on detail, max 20 tokens>, "structure":<comment on
       structure, max 20 tokens> },
       "revision":  <improved version of the answer, max 2x tokens of input if > 2 tokens,
       otherwise max 20 tokens> }
```

Figure 5: Instructions provided to the system for analyzing the quality of answers based on different criteria and generating a revised response in JSON format.

Table 14: Dataset statistics for 3 medical VQA datasets: VQA-RAD, SLAKE, and PathVQA.

| Dataset | VQA-RAD | | SLAKE | | | PathVQA | | |
|---|---|---|---|---|---|---|---|---|
| | Train | Test | Train | Val | Test | Train | Val | Test |
| # Images | 313 | 203 | 450 | 96 | 96 | 2599 | 858 | 858 |
| # QA Pairs | 1797 | 451 | 4919 | 1053 | 1061 | 19755 | 6279 | 6761 |
| # Open | 770 | 179 | 2976 | 631 | 645 | 9949 | 3144 | 3370 |
| # Closed | 1027 | 272 | 1943 | 422 | 416 | 9806 | 3135 | 3391 |

## H.2 Results

Tables 15 and 16 present the results using 70% and 100% of the data. Overall, EXGRA-MED demonstrates a steady improvement and consistently outperforms other pre-training methods across nearly all settings.

Table 15: Performance fine-tuning on MedVQA downstream datasets (pre-training 70%). **Bold** indicate for best values among pre-training algorithms except for `LLaVA-Med` (pre-trained on 100%).

| Method | VQA-RAD | | | SLAKE | | | PathVQA | | | Overall |
|---|---|---|---|---|---|---|---|---|---|---|
| | Open | Closed | Avg. | Open | Closed | Avg. | Open | Closed | Avg. | |
| LLaVA-Med (100%) | 63.65 | 81.62 | 72.64 | 83.44 | 83.41 | 83.43 | 36.78 | 91.33 | 64.06 | 73.37 |
| LLaVA-Med (70%) | 65.96↑2.31 | 81.62↓0 | 73.79↑1.13 | 84.16↑0.72 | 83.17↓0.24 | 83.67↑0.24 | 37.39↑0.61 | 92.27↑0.94 | 64.83↑0.77 | 74.1↑0.64 |
| InfoNCE | 64.18 | 77.94 | 71.06 | 70.9 | 82.69 | 76.80 | 33.58 | 88.5 | 61.04 | 69.63 |
| PLOT | 60.13 | 78.31 | 69.22 | 82.48 | 83.89 | 83.185 | 29.23 | 85.7 | 57.478 | 69.96 |
| SigLIP | 61.68 | 78.68 | 70.18 | 82.04 | 83.17 | 82.61 | 34.43 | 90.3 | 62.37 | 71.72 |
| VLAP | 64.08 | 79.41 | 71.75 | **84.94** | **85.1** | **85.02** | 36.44 | 91.51 | 63.98 | 73.58 |
| **ExGra-Med** | **67.12** | **81.99** | **74.56** | 84.81 | 84.86 | 84.84 | 37.26 | 91.77 | 64.52 | **74.64** |

Table 16: Performance fine-tuning on MedVQA downstream datasets (pre-training 100%).

| Method | VQA-RAD | | | SLAKE | | | PathVQA | | | Overall |
|---|---|---|---|---|---|---|---|---|---|---|
| | Open | Closed | Avg. | Open | Closed | Avg. | Open | Closed | Avg. | |
| LLaVA-Med (100%) | 63.65 | 81.62 | 72.64 | 83.44 | 83.41 | 83.43 | 36.78 | **91.33** | **64.06** | 73.37 |
| InfoNCE | 66.01 | 79.41 | 72.71 | 83.23 | 83.41 | 83.32 | 35.01 | 89.53 | 62.27 | 72.77 |
| PLOT | 63.58 | 77.21 | 70.4 | 82.44 | 84.86 | 83.65 | 34.45 | 89.97 | 62.21 | 72.09 |
| SigLIP | 57.11 | 74.26 | 65.69 | 85.07 | 83.41 | 84.24 | 36.47 | 89.38 | 62.925 | 70.95 |
| VLAP | 60.93 | 79.78 | 70.36 | 84.74 | 83.17 | 83.955 | 35.86 | 89.65 | 62.755 | 72.36 |
| **ExGra-Med** | **66.35** | **83.46** | **74.91** | **85.34** | **85.58** | **85.46** | **36.82** | 90.92 | 63.87 | **74.75** |

# I  Further Ablation Studies

## I.1  $K$ Nearest Neighbor in the Graph Construction Step

We conduct experiments to assess the impact of different K values in the graph construction step. Table 17 presents model performance on the VQA-RAD dataset along with the training time for Step-2 pre-training using 10% of the data for each $K$ value. Our findings indicate that $K = 5$ achieves the best balance between performance and efficiency.

Table 17: Impact of Nearest Neighbors Count on Graph Construction. Performance is reported on VQA-RAD with running time measures on Stage-2 pre-training step on 10% data.

| Settings | VQA-RAD | | | |
|---|---|---|---|---|
| | Open | Close | Avg. | Run Time |
| ExGra-Med (Full), K = 3 | 55.9 | 73.9 | 64.9 | 1h |
| ExGra-Med (Full), K = 5 | 66 | 79.04 | 72.52 | 1h4' |
| ExGra-Med (Full), K = 7 | 55.52 | 73.16 | 64.37 | 1h17' |

Table 18: Comparison of pre-training algorithms with different feature embedding methods. Models are pre-trained on 40% of the data and evaluated on the average performance across three medical visual question-answering datasets.

| Method | VQA-RAD | SLAKE | PathVQA |
|---|---|---|---|
| ExGra-Med | **74.37** | **84.99** | **64.34** |
| InfoNCE (avg.feature) | 70.34 | 83.29 | 61.6 |
| PLOT (optimal transport) | 71.89 | 83.16 | 62.4 |

## I.2  Feature representation analysis using average pooling for visual and language tokens

We investigate using average pooling token features in ExGra-Med with two experiments:

- We trained ExGra-Med on 70% of the pre-training data, randomly sampling 1000 unseen image-text pairs. The trained model extracted features using average pooling, and a box plot (Figure 6) visualized the central tendency, spread, and skewness of 1000 positive and negative pairs. The results show: (i) the median similarity for positive pairs is significantly higher than for negative pairs, indicating clear separation; (ii) while some overlap exists in the interquartile ranges (IQRs), the shift in central tendency confirms the distinction; and (iii) outliers are present, particularly among negative pairs, but they minimally overlap with the core distribution of positive pairs.

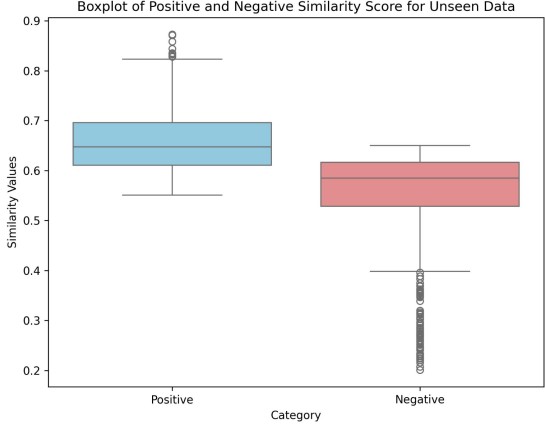

Figure 6: Visualization of similarity values between positive and negative pairs based on features computed by ExGra-Med.

- We compare against two pre-training algorithms, InfoNCE [45, 58] and PLOT [15]. Both utilize the same contrastive loss, but InfoNCE relies on *cosine distance with averaged features*, while PLOT directly *computes optimal transport over sets of visual and text tokens*. The results for these baselines are summarized in Table 18. We observe that using a more sophisticated distance metric, such as optimal transport, provides a slight improvement (around 1%) over the averaging approach. However, the performance gain is relatively modest. Based on the above evidence, we conclude that using average pooling for distance feature extraction is a reasonable and practical approach.

## J    Qualification Test on the GPT-generated Extended Captions

We adopt the GPT-4 as a tool for paraphrasing image captioning due to its improved performance compared with GPT-3.5, especially in healthcare [40]. During our implementations, we also randomly checked for a hundred samples and found consistency between extended context and original ones. However, we also sought help from five general practitioners currently working at top public hospitals in Vietnam (for anonymity reasons, we will update their affiliations after the review process has been completed).

In particular, we randomly chose 1000 samples in Stage 2 of pre-training, covering five data modalities: chest X-ray, CT scan, MRI, histology, and others. Each doctor is assigned a specific data modality given their expertise, including 200 image-text pairs and corresponding captions. We then build an annotation tool for them to verify data where each sample is asked with two questions (i) whether the extended caption covers the original caption; and (ii) whether new concepts appearing in extended captions are correct. For (i) and (ii), doctors can rate with five levels (from 1 to 5), each indicating an increasing level of correctness (Figures 7-8).

We provide statistical correctness evaluated by general doctors for these domains in Figures 10,11,12, 13, and 14. It can be seen that most rating scores fall between 3 and 5, with only a small number of samples rated 1 or 2, validating the overall consistency of GPT-4 outputs. While concerns may arise regarding the impact of low-scoring extended captions (rated 1 or 2) on the LLM, it's important to note that these extended captions are utilized solely for contrastive learning during pre-training to align the model's latent space representations. They are not used in auto-regressive training, which involves predicting target ground-truth tokens. Additionally, the model is fine-tuned with the given training samples from downstream tasks after pre-training (no extended captions are used). Thus, we argue that the presence of a small number of noisy extended captions should not significantly affect the performance of the LLM.

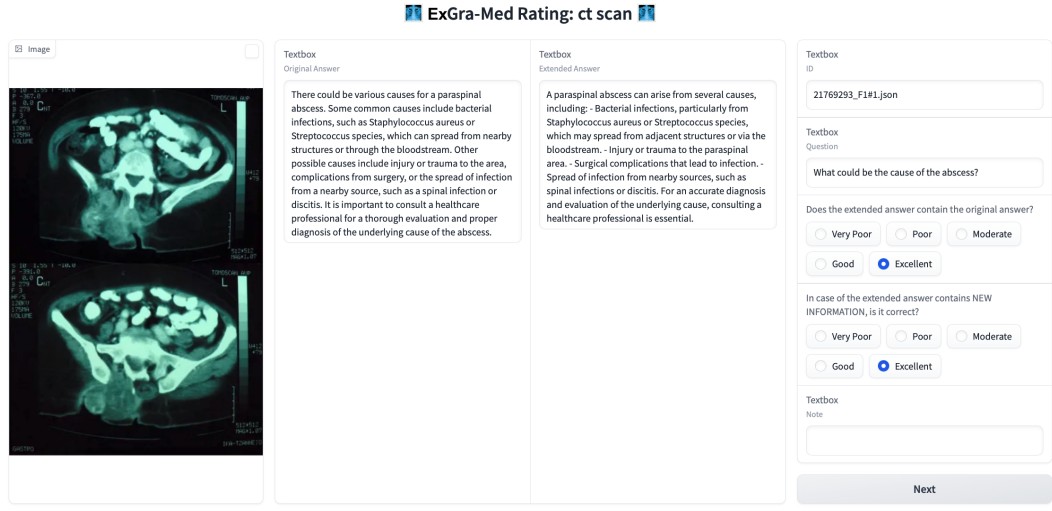

Figure 7: (Part 1) Demonstration of our annotation tool for general practitioners to validate the quality of extended captions generated by GPT-4.

**LoGra-Med Rating Guidelines**

**1: Very Poor Consistency**

○ **Description**: The extended caption significantly diverges from the original meaning, includes incorrect or irrelevant medical information, or introduces significant factual errors.

○ **Examples**:

○ Contradicts the original caption.

○ Adds details that are medically implausible or incorrect.

○ Does not maintain any coherence with the original content.

**2: Poor Consistency**

○ **Description**: The extended caption retains some elements of the original caption but includes inaccuracies or overly speculative content. The new details are loosely related to the original or contextually irrelevant.

○ **Examples**:

○ Partial preservation of the original meaning.

○ Contains minor factual errors or misinterpretations.

○ Unnecessarily diverges into unrelated topics.

**3: Moderate Consistency**

○ **Description**: The extended caption mostly aligns with the original caption but includes minor inaccuracies, redundant information, or slightly irrelevant expansions. The medical context remains intact but could be improved.

○ **Examples**:

○ Retains the main idea but adds unnecessary or repetitive details.

○ Expansion is overly generic and lacks depth in medical relevance.

**4: Good Consistency**

○ **Description**: The extended caption is well-aligned with the original caption, provides accurate and relevant additional details, and enhances the context without deviating from the medical focus.

○ **Examples**:

○ Adds valuable, medically relevant information that complements the original.

○ Maintains high factual accuracy and stays within the context.

**5: Excellent Consistency**

○ **Description**: The extended caption perfectly aligns with the original, enriching the content with precise, relevant, and insightful details. It enhances understanding without introducing errors or deviating from the topic.

○ **Examples**:

○ Seamlessly extends the original caption with meaningful medical context.

○ Fully accurate and maintains clarity and relevance.

Figure 8: (Part 2) A detailed guideline for scoring, ranging from 1 to 5, is provided.

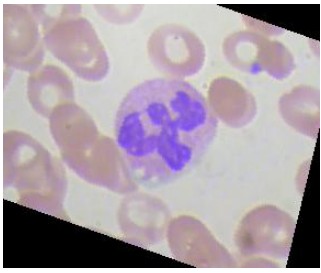

Q: What are the types of cells depicted in this image?

A: Neutrophils

B: Melanocytes
C: Lymphocytes
D: Hepatocytes

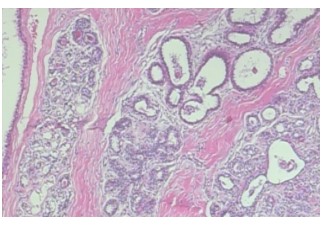

Q: What is the diagnosis of the histopathology in this image?
A: Breast hyperplasia without atypia histopathology
B: Normal breast histopathology

C: Benign breast histopathology

D: Fibrocystic breast histopathology

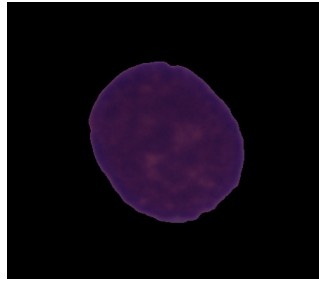

Q: What is the probable diagnosis depicted in this image?
A: Chronic myeloid leukemia
B: Multiple myeloma
C: Hodgkin's lymphoma

D: Acute lymphoblastic leukemia

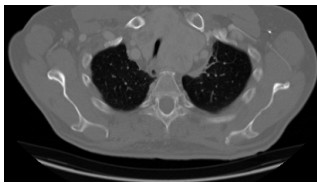

Q: What is the diagnosis of the cancer seen in this image?
A: Adenocarcinoma of the right hilum, T3 N1 M0, Stage IIb
B: Mesothelioma of the right hilum, T2 N1 M0, Stage IIb

C: Large cell carcinoma of the left hilum, T2 N2 M0, Stage IIIa

D: Non-small cell carcinoma of the left hilum, T2 N0 M0, Stage I

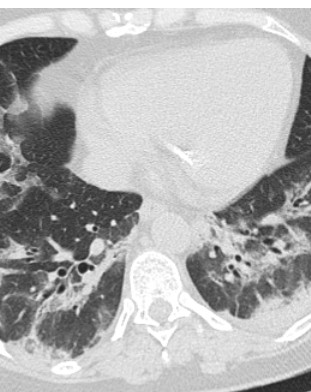

Q: Is COVID-19 apparent in this CT scan image?
A: No

B: Yes

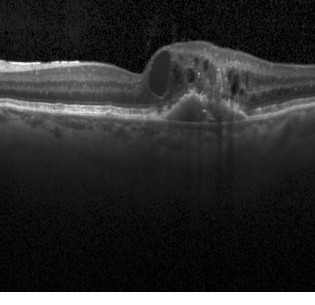

Q: Which imaging technique was utilized to obtain this image?
A: Ultrasound

B: Optical Coherence Tomography

C: Magnetic Resonance Imaging (MRI)
D: Thermography

Figure 9: Examples from the OmniMedVQA dataset: microscopy (top) and CT images (bottom) with corresponding questions and options, with the correct answers highlighted in blue.

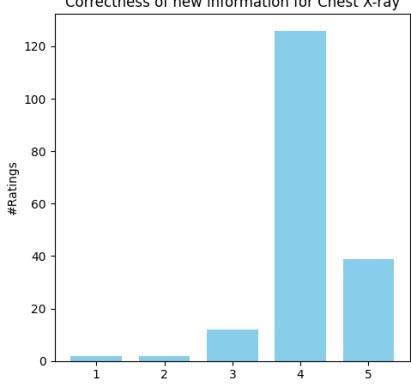

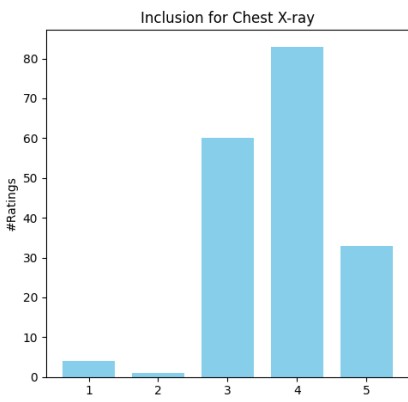

Figure 10: Statistical correctness of extended captions generated by GPT-4 on Chest X-rays.

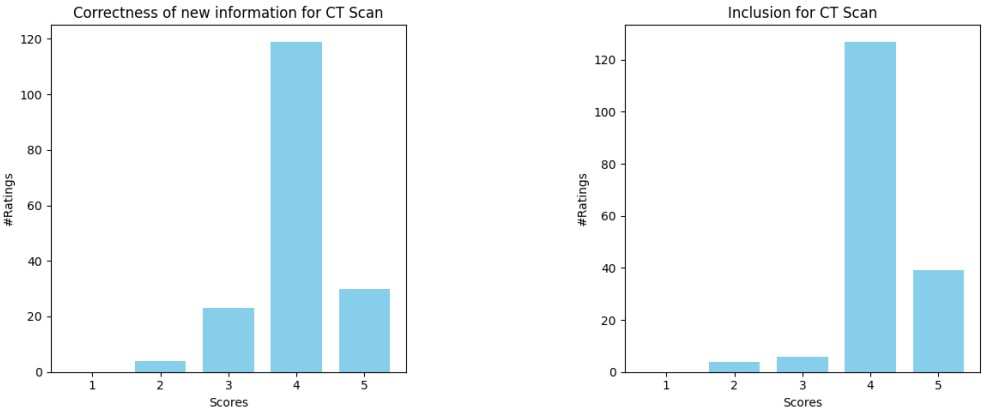

Figure 11: Statistical correctness of extended captions generated by GPT-4 on CT scans.

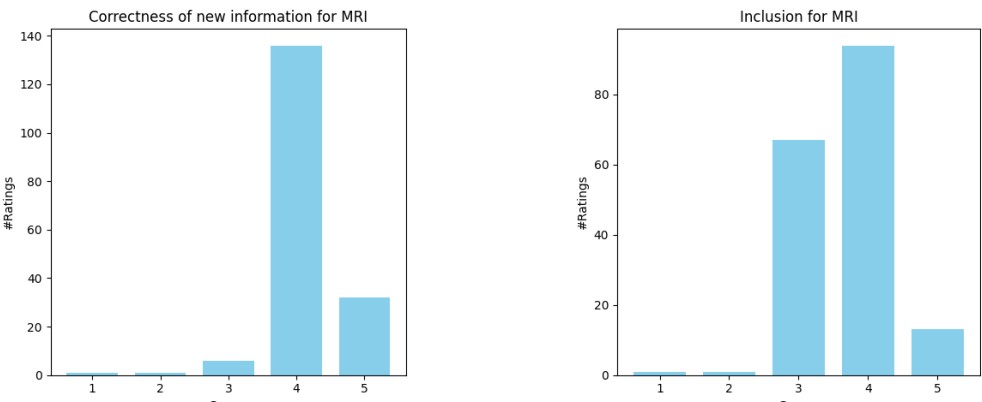

Figure 12: Statistical correctness of extended captions generated by GPT-4 on MRI.

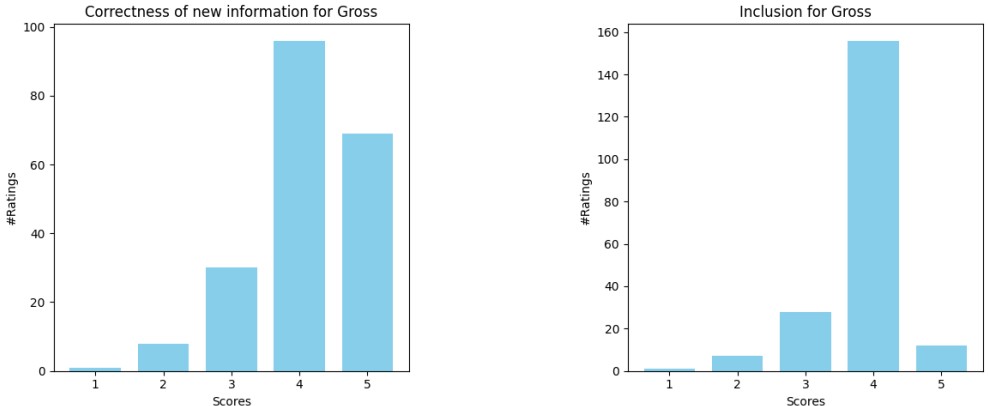

Figure 13: Statistical correctness of extended captions generated by GPT-4 on mixed domains.

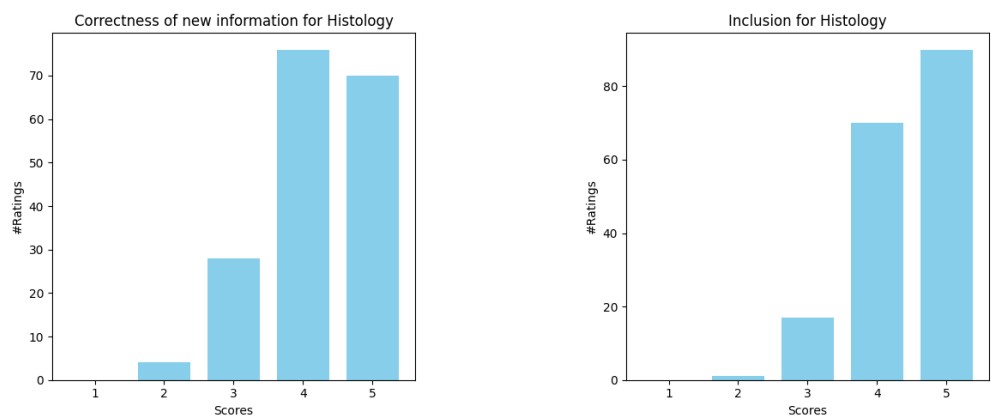

Figure 14: Statistical correctness of extended captions generated by GPT-4 on histology samples.

