# OpenReview forum: "ExGra-Med: Extended Context Graph Alignment for Medical Vision-Language Models"
_NeurIPS.cc/2025/Conference — NeurIPS 2025 poster_

### Official Review · Reviewer_H5Kb · 2025-06-15

**Clarity:** 3
**Significance:** 4
**Originality:** 3
**Rating:** 4
**Confidence:** 3

**Summary:**

Existing medical multimodal LLMs rely heavily on large-scale data and autoregressive training, which often results in poor vision-language alignment and high dependence on instruction-tuning data. The authors propose ExGra-Med, a novel multi-graph alignment framework that jointly aligns images, responses, and extended captions to enhance semantic grounding and cross-modal coherence. ExGra-Med achieves competitive or superior performance to state-of-the-art models like LLaVA-Med and BioMedGPT using significantly less training data, showing strong efficiency and scalability for medical vision-language tasks.

**Questions:**

See Weaknesses.

**Ethical Concerns:**

["NO or VERY MINOR ethics concerns only"]

**Final Justification:**

The author's response has addressed some of my concerns.

**Limitations:**

Yes.

**Paper Formatting Concerns:**

No.

**Quality:**

4

**Strengths And Weaknesses:**

Pros:

1. The paper identifies a critical limitation in existing autoregressive medical MLLMs by revealing their heavy reliance on large-scale instruction-following data, highlighting a key inefficiency in current pre-training paradigms.

2. It introduces a novel multi-graph alignment objective that effectively captures triplet-level semantic relationships among images, instruction responses, and enriched captions, enhancing cross-modal understanding.

3. An efficient training solver is proposed for large LLMs, supported by theoretical insights into geodesic distances and shortest paths in multimodal graph space.

4. The proposed model, EXGRA-MED, achieves competitive or superior performance to state-of-the-art med-MLLMs using only a fraction of the data, demonstrating strong efficiency, scalability, and generalization across diverse medical vision-language tasks.

Cons:

1. Does EXGRA-MED use full fine-tuning? How does it perform with parameter-efficient fine-tuning methods such as LoRA, and could this further improve efficiency?

2. The proposed method appears to be domain-agnostic---could it also be effective in the general domain beyond the medical field?

3. It would be helpful to cite a recent survey on MLLMs to provide readers with a comprehensive background and better contextualize your contribution within the broader MLLM research landscape [1].

[1] "MM-LLMs: Recent Advances in MultiModal Large Language Models." Findings of the Association for Computational Linguistics ACL 2024. 2024.

---

> ### Author Rebuttal · Authors · 2025-07-31
>
> We sincerely thank the reviewer for their thoughtful and supportive feedback. We also appreciate your positive evaluation of the core contributions of our work, including identifying the limitations of current medical MLLMs, introducing the multi-graph alignment objective, and proposing an efficient solver grounded in theoretical insights. Below we address your remaining concerns.
>
> **Q1. Does EXGRA-MED use full fine-tuning? How does it perform with parameter-efficient fine-tuning methods such as LoRA, and could this further improve efficiency?**
>
> **A1** We thank the reviewer for this insightful question. To clarify, all experiments in the main paper **use full fine-tuning** to ensure fair comparisons with existing baselines, which are also fully fine-tuned.
>
> To investigate the reviewer’s suggestion, we conducted additional experiments using **LoRA-based parameter-efficient fine-tuning** on both **ExGra-Med** and **LLaVA-Med**, and evaluated their performance on the VQA-RAD and SLAKE datasets. The results are summarized below:
>
> | Method                          | VQA-RAD Open | VQA-RAD Closed | VQA-RAD Avg | Param |
> |---------------------------------|--------------|----------------|-------------|--------|
> | ExGra-Med (LoRa)               | 63.55        | 79.41          | 71.48       | 2.1B   |
> | LLaVa-Med (LoRa)               | 62.06        | 75.00          | 68.53       | 2.1B       |
> | ExGra-Med (Fully-finetuning)   | 66.35        | 83.46          | 74.91       | 7B     |
> | LLaVa-Med (Full-finetuning)    | 63.65        | 81.62          | 72.64       | 7B     |
>
> These results show that **ExGra-Med with LoRA still outperforms LLaVA-Med with LoRA**, indicating the robustness of our approach even under parameter-efficient tuning. However, as expected, **LoRA-based tuning does lead to a drop in performance compared to full fine-tuning**.
>
> We believe that to bridge this gap, **pre-training on a larger-scale medical instruction-tuning dataset** (e.g., the 21M samples curated from MedTrinity [1]) would allow the LoRA setup to more closely match the performance of fully fine-tuned models. If the paper is accepted, we plan to explore this direction further and release a variety of pretrained checkpoints, including LoRA-compatible versions, to support the community.
>
> [1] Xie, Yunfei, et al. "MedTrinity-25M: A Large-Scale Multimodal Dataset with Multigranular Annotations for Medicine." ICLR 2024.
>
> **Q2. The proposed method appears to be domain-agnostic - could it also be effective in the general domain beyond the medical field?**
>
> **A2**.  This is a very insightful question. We thank you for raising it. While our starting focus in this work is on medical multimodal LLMs, we highlight a broader challenge: autoregressive models tend to be data-hungry, especially when adapting across domain shifts (e.g., from natural images to medical images).
>
> We believe this issue might not be limited to the medical domain. **For instance, in the robotics domain**, particularly in vision-language-action (VLA) models [1,2], similar challenges can arise. These models often face limited access to high-quality training trajectories and require significant adaptation when deployed in complex real-world environments. As a result, they may demand large-scale pretraining to bridge the domain gap.
>
> Overall, we see this as an important direction for future research and believe our proposed multi-graph alignment approach could generalize well to such domains, potentially improving efficiency and cross-modal alignment where labeled data is scarce.
>
> [1] Kim, Moo Jin, et al. "OpenVLA: An open-source vision-language-action model." Arxiv 2024
>
> [2] Black, Kevin, et al. "π0: A vision-language-action flow model for general robot control. Arxiv 2024
>
> **Q3. Adding a more recent survey on MLLMs to provide readers with a comprehensive background and better contextualize your contribution within the broader MLLM research landscape.**
>
> **A3**. We thank the reviewer for this valuable suggestion. We agree that incorporating recent surveys on medical multimodal large language models (MLLMs) would strengthen the contextual grounding of our work. In the revised version, we will update the Introduction and Related Work sections to include relevant and up-to-date surveys, thereby better situating our contributions within the evolving MLLM research landscape.

---

> > ### Comment · Reviewer_H5Kb · 2025-08-04
> >
> > Thanks for your response. I decided to keep my positive rating.

---

> > > ### Author Response · Authors · 2025-08-09
> > > **Thank you Reviewer H5Kb**
> > >
> > > We sincerely thank Reviewer H5Kb once again for your feedback and for continuing to view our work positively.

---

### Official Review · Reviewer_xkmU · 2025-06-20

**Clarity:** 4
**Significance:** 4
**Originality:** 4
**Rating:** 5
**Confidence:** 5

**Summary:**

This paper proposes EXGRA-MED, a training-efficient framework for medical vision-language models that aligns images, instructions, and enriched captions through a multi-graph alignment strategy. By combining this with black-box optimization, the model achieves strong performance with significantly less instruction-following data, outperforming baselines across multiple medical VQA and classification tasks.

**Questions:**

- Could the authors provide quantitative results when training EXGRA-MED with smaller frozen LLMs to evaluate downstream performance trade-offs?

- How robust is EXGRA-MED to inaccurate or noisy extended captions, especially if GPT-4 is replaced with lower-quality generators?

- Is the triplet graph structure essential, or could similar benefits be achieved using just pairs (e.g., image and extended caption)?

- Can the authors release a toolkit or provide open-source code for the black-box optimization with IMLE to support reproducibility?

**Ethical Concerns:**

["NO or VERY MINOR ethics concerns only"]

**Final Justification:**

Thanks for the rebuttal. It solves all the concerns.

**Limitations:**

No. Authors should be rewarded rather than punished for being up front about the limitations of their work and any potential negative societal impact.

**Quality:**

4

**Strengths And Weaknesses:**

Strengths
- The paper introduces a multi-graph alignment framework that directly trains LLMs with triplet correlations across images, instruction responses, and extended captions, improving vision-language coherence without requiring large datasets
- EXGRA-MED demonstrates strong performance using only 10% of the instruction-following data, reaching or exceeding LLaVA-Med's full-data performance on multiple benchmarks.
- Employs Implicit Maximum Likelihood Estimation to scale combinatorial graph alignment efficiently to LLaMA-7B-sized models, a notable technical achievement.
- Covers three major MedVQA datasets, chatbot benchmarks, and zero-shot classification on 23 datasets, showing consistent improvements over several baselines.

Weaknesses
- While LLaVA-Med and LLaMA-7B are used as backbones, the method’s generalization to other base models (e.g., Qwen) remains underexplored.
- The enriched caption generation pipeline depends on proprietary GPT-4 APIs. This reliance may hinder reproducibility or deployment in restricted environments.
- The graph alignment formulation, though elegant, is mathematically intensive and may benefit from simplified descriptions or more intuitive illustrations for accessibility.

---

> ### Author Rebuttal · Authors · 2025-07-31
>
> We sincerely thank the reviewer for the thoughtful and encouraging feedback. We greatly appreciate your recognition of the key contributions of our work, including the design of the multi-graph alignment framework, the efficient use of limited instruction-following data, and the successful integration of IMLE for scalable graph alignment. Below we address Reviewers’ concerns.
>
>
> **Q1. Examine ExGra-Med’s generalization to other base models (e.g. Qwen).**
>
> **A1**. We thank the Reviewer for your suggestion. During the short period of rebuttal, we integrated the ExGrad-Med to the Qwen-VL 2.5, 7B version. This model is pre-trained with the original 10% medical instruction-tuning data and the extended version with multi-graph alignments. We summarize below our findings confirming that ExGra-Med still helps to enhance the Qwen model in this case. We’ll update this setting with 100% pre-training to the revised version if the paper is accepted.
>
> | Model                                           | VQA-RAD | SLAKE  |
> |------------------------------------------------|---------|--------|
> | Qwen-VL (general weights)                      | 53.85%  | 70.5%  |
> | Qwen-VL Med (medical pre-trained 10%)          | 52.73%  | 86.53% |
> | ExGra-Med (Qwen-VL, pre-trained 10%)           | 58.10%  | 89.75%    |
>
>
> **Q2. Using proprietary GPT-4 APIs to generate enriched captions may hinder reproducibility or deployment in restricted environments.**
>
> **A2** We thank the reviewer for highlighting these important concerns related to **reproducibility** and **practical deployment**.
>
> To support **reproducibility**, we will publicly release the GPT-4, generated extended instruction-following dataset used in our experiments. This will allow researchers to reproduce our results without needing API access.
>
> **Regarding deployment in restricted environments**, for example, in healthcare settings with strict data privacy constraints where sending data to external APIs (like GPT-4) may not be permissible, we address this by validating alternative caption generation using open-source models. Specifically, we employ the Qwen3-8B LLM [1], which has demonstrated superior performance compared to previous Qwen variants (e.g., QwQ, Qwen2.5) on reasoning-intensive tasks like mathematics, coding, and commonsense reasoning.
>
> Due to the time constraints of the rebuttal phase, we were only able to evaluate ExGra-Med pre-trained on 10% Qwen-generated instruction-following data. We summarize the obtained results in the table below
>
> | Model                                  | VQA-RAD Open | VQA-RAD Closed | VQA-RAD Avg | SLAKE Open | SLAKE Closed | SLAKE Avg |
> |----------------------------------------|--------------|----------------|-------------|------------|--------------|-----------|
> | ExGra-Med (Qwen-enhanced captions)     | 61.96        | 78.31          | 70.13       | 82.6       | 86.8         | 84.7      |
> | ExGra-Med (GPT-4 enhanced captions)    | 66.02        | 79.04          | 72.52       | 84.92      | 85.10        | 85.01     |
> | LLaVA-Med (baseline)                   | 43.38        | 61.4           | 52.39       | 80.94      | 80.29        | 80.62     |
>
> As can be seen that, ExGra-Med (Qwen-enhanced captions) achieves:
>
> **+17.74%** higher average accuracy on **VQA-RAD** (70.13 vs. 52.39),
>
> **+4.08%** higher average accuracy on **SLAKE** (84.7 vs. 80.62), compared to the LLaVA-Med baseline.
>
> While the performance of ExGra-Med with GPT-4-enhanced captions is notably higher on VQA-RAD (72.52 vs. 70.13), the two models achieve comparable results on SLAKE (85.01 vs. 84.7), suggesting that our multi-graph alignment remains effective even when using alternative instruction-following data sources.
> We will add these findings more explicitly in the revised manuscript if the paper is accepted.
>
> [1] Yang, An, et al. "Qwen3 technical report." Arxiv 2025
>
> **Q3. The graph formulation is mathematically intensive and may benefit from simplified descriptions.**
>
> **A3**. We thank the reviewer for this valuable suggestion. We agree that the current mathematical formulation may be too technical and could benefit from a more accessible presentation to better reach a broader audience. If the paper is accepted, we will revise the text to simplify the mathematical descriptions and provide more intuitive explanations, including visual illustrations and video teasers on the project website, to make the core ideas of the method easier to understand and follow.
>
> **Q4. How can ExGra-Med remain robust when replacing GPT-4 with potentially lower-quality caption generators?**
>
> **A4**. We appreciate the reviewer’s question and have addressed this concern in conjunction with Q2. Specifically, we evaluated the robustness of ExGra-Med by replacing GPT-4 with a state-of-the-art open-source model, Qwen3-8B, to generate extended instruction-following captions.
>
> As reported, even when using captions generated by Qwen3-8B, arguably of lower quality than GPT-4, ExGra-Med still achieves a significant improvement over the LLaVA-Med baseline, confirming the robustness of our multi-graph alignment approach to variation in caption quality.
>
> **Q5. Is the triplet graph structure essential, or would simple pairwise alignments between image and extended captions suffice?**
>
> **A5**. We thank the reviewer for this insightful question. We had the same curiosity and conducted an **ablation study** to investigate the necessity of the triplet graph structure.
>
> In this study, we compared the full **triplet alignment** setup (image, original caption, and extended caption) against a simplified version using only **pairwise alignments** (image and extended caption). Both models were pre-trained using **40% of the medical instruction-tuning data**, and results are shown in Table 6 of the main paper. For convenience, we summarize the key results below:
>
> | Method                                  | VQA-RAD | SLAKE  |
> |-----------------------------------------|---------|--------|
> | ExGra-Med (triplet alignment)           | 74.37   | 84.99  |
> | ExGra-Med (pairwise alignment only)     | 72.58   | 82.31  |
>
> These results indicate that the **triplet alignment structure provides a meaningful improvement**, likely due to its ability to better capture the semantic relationships across all three modalities. This richer structure helps the model learn more robust and informative representations compared to simpler pairwise alignments.
>
> **Q6. Can the authors release a toolkit or provide open-source code for the black-box optimization with IMLE to support reproducibility?**
>
> **A6**. Yes, we fully support open science and reproducibility. Upon acceptance, we will release:
> - **The pretrained checkpoints** (based on multi-graph alignment) for downstream task adaptation, and
>
> - **The complete training code**, including our efficient solver with barycenter graph construction and end-to-end learning using black-box gradient estimation via IMLE.
>
> This will enable the community to reproduce our results from scratch, starting from the official LLaVA checkpoints, and facilitate further research based on our framework.
>
> **Q7. Quantitative results when training EXGRA-MED with smaller frozen LLMs to evaluate downstream performance trade-offs**
>
> **A7**. We thank the reviewer for this thoughtful suggestion. We fully agree that evaluating ExGra-Med with smaller language models would provide valuable insight into the trade-off between model size, efficiency, and performance.
>
> Due to the limited timeframe of the rebuttal period, we were unable to conduct additional experiments with significantly smaller models (e.g., 3B or below). However, we partially addressed this concern by validating the robustness of **ExGra-Med on the Qwen-VL 7B model, as discussed in Answer 1**. This evaluation demonstrates that ExGra-Med generalizes well beyond LLaVA-Med and retains its performance gains when applied to a different backbone.
>
> We recognize the importance of further exploring the performance of ExGra-Med on smaller frozen LLMs to better support deployment in resource-constrained settings. If the paper is accepted, **we plan to extend our analysis to smaller models, such as Qwen-VL 3B or MiniGPT variants**, and release the corresponding checkpoints and results to benefit the community.

---

> > ### Comment · Reviewer_xkmU · 2025-08-05
> > **Thanks for your response. I decided to keep my positive rating.**
> >
> > Thanks for your response. I decided to keep my positive rating.

---

> > > ### Author Response · Authors · 2025-08-09
> > > **Thank you Reviewer xkmU**
> > >
> > > Before concluding the discussion phase, we would like to thank the Reviewer for their time and thoughtful feedback, as well as for maintaining their positive evaluation of our work.

---

### Official Review · Reviewer_Fpu5 · 2025-07-02

**Clarity:** 3
**Significance:** 2
**Originality:** 3
**Rating:** 4
**Confidence:** 3

**Summary:**

This paper introduces EXGRA-MED, a novel multi-graph alignment framework for medical vision-language models that addresses the data-hungry nature of autoregressive training in medical multimodal LLMs. The authors identify that models like LLaVA-Med suffer significant performance drops when trained on limited instruction-following data, even after fine-tuning. To address this, EXGRA-MED constructs three modality-specific graphs representing visual features, original instruction responses, and GPT-4 generated extended captions, then jointly aligns them using a barycenter graph approach with black-box gradient estimation for scalable training.

**Questions:**

1. You mention the ExGra-Med's language model backbone is LLaMA-7B, but which LLaMA version exactly? From what I know, LLaVA-med uses mistral-7b-v0.2 as the language model backbone. Usually better language model backbone will result in better downsteam performance, will that cause unfair comparison?
2. Please try using the term "pre-training" for stage 1 and "fine-tuning" on stage 2 properly, according to the original LLaVA paper. On line 281, I assume you want to say "complete the whole fine-tuning process compared to the LLaVa-Med".

**Ethical Concerns:**

["NO or VERY MINOR ethics concerns only"]

**Final Justification:**

The rebuttal solves my concerns

**Limitations:**

yes

**Quality:**

3

**Strengths And Weaknesses:**

## Strengths
1. The paper clearly demonstrates the data-hungry nature of autoregressive training in medical MLLMs through systematic experiments showing dramatic performance drops (e.g., 72.64% to 52.39% on VQA-RAD) when using only 10% of pre-training data.
2. The novel fine-tuning with multi-graph alignment

## Weaknesses
1. Training efficiency analysis. According to section 4.1 the actual training time of ExGra-med is 0.5 hour longer than the original LLaVA-Med. How much additional cost (e.g. GNN training) will the multi-graph alignment training will bring should be clarified.

---

> ### Author Rebuttal · Authors · 2025-07-31
>
> We thank the reviewer for the encouraging feedback and thoughtful observations regarding both the strengths and areas for improvement in our work. Regarding the training efficiency concern:
>
> **Q1. Clarifying additional costs of 0.5 hour longer of ExGraMed compared to LLavA-Med.**
>
> **A1**. We thank the reviewer for raising this important point. The **additional 0.5-hour training time** required by ExGra-Med compared to LLaVA-Med is primarily **due to the multi-graph alignment module**. This includes the following computational steps:
>
> - Constructing graphs using the **K-nearest neighbors** algorithm,
>
> - Performing **GNN-based message passing** to aggregate features among nodes in each graph,
>
> - Solving **three graph alignment** problems during the forward pass using a barycenter graph, and
>
> - Executing the corresponding **backward steps** to compute gradients for optimization.
>
> To ensure that the observed performance gain of ExGra-Med is not simply due to longer training time, we conducted an additional ablation experiment where **LLaVA-Med was trained for one extra hour**. As shown in **Table 3**, ExGra-Med still outperforms LLaVA-Med under these settings, confirming the effectiveness of our proposed alignment method beyond just training duration.
>
> **Q2. Clarifying the language models LLAMA versions used in ExGra-Med and LLaVa-Med**
>
> **A2**. Thank you for your careful evaluation and valuable question. To ensure a fair comparison, we confirm that both **ExGra-Med** and **LLaVA-Med** in our study are built upon the **same original LLaMA-7B model**, as provided on Hugging Face [1]. Specifically, **we do not use** any variant such as the **Mixture-of-Experts** mistral-7B-v0.2 checkpoint. This ensures consistency and fairness across all experiments.
>
> [1] huggyllama/llama-7b
>
> **Q3. Make consistent use of the terms “pre-training” and “fine-tuning”**
>
> **A3**. Thank you for the helpful comment. We acknowledge the potential confusion caused by the use of the terms “pre-training” and “fine-tuning.”
>
> Following the original LLaVA paper (Figure 3), Stage 1 is intended for the medical concept alignment, and Stage 2 is used for fine-tuning on generic medical instruction tuning. **However, both these Stages indeed can be seen as the pre-training phase** because the model is still not fully optimized for specific downstream tasks. **Additional fine-tuning** is required on each target dataset (e.g., VQA-RAD, SLAKE, or PathVQA) to achieve the best performance.
>
> Regarding line 281, we intended to state that ExGra-med takes slightly longer to complete the full pre-training process (Stage 1 + Stage 2) on the 600k medical instruction tuning samples compared to LLaVA-Med.
>
> We will revise this sentence for clarity and ensure consistent use of terminology in line with the original LLaVA definitions throughout the manuscript.

---

### Official Review · Reviewer_Usn8 · 2025-07-03

**Clarity:** 2
**Significance:** 3
**Originality:** 3
**Rating:** 3
**Confidence:** 3

**Summary:**

This paper presents ExGRA-Med, a multi-graph alignment framework designed to improve vision-language alignment in medical multimodal large language models. The authors address a significant limitation of existing approaches (e.g., LLaVA-Med, BioMedGPT) that suffer from weak cross-modal alignment due to their heavy reliance on autoregressive training objectives.

**Questions:**

1. The proposed method uses GPT-4o-powered extended context instruction-following data for multi-graph alignment. How does this compare with methods that directly optimize instruction-following data using GPT-4o?  Direct comparison with GPT-4o optimization methods would strengthen contribution claims.
2. Current medical VQA datasets are typically more complex than the Figure 2 examples. Can your method maintain effectiveness on sophisticated medical datasets involving complex clinical reasoning? Demonstrating effectiveness in complex medical scenarios would enhance practical significance.
3. Please incorporate more challenging benchmarks such as MMMU Health & Medicine, MedXpertQA, and OmnimedVQA to better assess model capabilities. Results on advanced benchmarks would provide stronger evidence of superiority.
Limitations
4. The complex alignment architecture requires more thorough validation beyond simple benchmarks. Effectiveness has not been sufficiently demonstrated across diverse scenarios.
5. Unclear whether performance gains persist with stronger base models or comprehensive VQA corpora. Robustness to state-of-the-art foundation models needs validation.

**Ethical Concerns:**

["NO or VERY MINOR ethics concerns only"]

**Limitations:**

1. The complex alignment architecture requires more thorough validation beyond simple benchmarks. Effectiveness has not been sufficiently demonstrated across diverse scenarios.
2. Unclear whether performance gains persist with stronger base models or comprehensive VQA corpora. Robustness to state-of-the-art foundation models needs validation.

**Quality:**

3

**Strengths And Weaknesses:**

Strengths:
1. The proposed multi-graph alignment mechanism effectively tackles the vision-language alignment problem by jointly aligning images, instruction responses, and extended captions in the latent space.
2. Demonstrates impressive data efficiency, achieving comparable performance to LLaVA-Med using only 10% of the pre-training data.
3. Shows substantial improvement (20.13% gain) on VQA-RAD benchmark, indicating better medical visual question answering capabilities.
4. The black-box gradient estimation approach enables scalable training for large LLMs.

Weakness:
1. The proposed multi-graph alignment framework introduces significant architectural complexity that makes the paper difficult to follow and understand. The intricate design may hinder reproducibility and practical adoption due to unclear implementation details and a lack of intuitive explanations.
2. The proposed approach extends VQA capabilities primarily through simple conversational formats as shown in Figure 2, limiting applicability to complex medical reasoning scenarios. The evaluation lacks assessment on more challenging conversational AI tasks that require sophisticated multi-turn dialogues or clinical decision-making processes.
3. The experimental evaluation is constrained to relatively simple benchmarks (VQA-RAD, SLAKE, PathVQA) and primarily compares against LLaVA-Med, an earlier generation model. The absence of comparisons with recent state-of-the-art medical multimodal models limits our understanding of the method's true competitive standing.

---

> ### Author Rebuttal · Authors · 2025-07-31
>
> We sincerely thank the reviewer for the thoughtful and detailed feedback. We greatly appreciate your recognition of the strengths of our work, including the effectiveness of the multi-graph alignment mechanism, the strong data efficiency, the significant gains on VQA-RAD, and the scalability enabled by our black-box optimization approach. We also appreciate the constructive suggestions regarding architectural clarity, evaluation scope, and broader applicability. Below, we address each of these concerns in detail.
>
> **Q1. Architectural complexity of multi-graph alignment, reproducibility, and practical adoption**
>
> **A1**. We thank the reviewer for highlighting these important aspects. Regarding the **architectural complexity** of the multi-graph alignment module, if the paper is accepted, we will simplify the mathematical formulations and include more intuitive explanations - such as **visual illustrations and video teasers** on the project website to make the method easier to understand and follow.
>
> In terms of **reproducibility** and **practical adoption**, we are committed to open-sourcing:
> - The **pretrained checkpoints** are used for downstream adaptation tasks, and
> - The **complete training pipeline** for multi-graph alignment
>
> This will ensure that our method can be fully reproduced and easily integrated by the research community.
>
> **Q2. Limitation of ExGra-Med’s conversational format in handling multi-turn dialogues or clinical decision-making**
>
> **A2**. We thank the reviewer for this question. We acknowledge that Figure 2 provides only a simplified illustration of how we extend the original captions. However, in practice, ExGra-Med is capable of handling multi-turn dialogues, as it is trained on curated multi-turn conversational data provided in the pre-training set of LLaVA-Med (Figure 5 in the LLaVA-Med paper).
>
> Specifically, for each input image, medical experts generate sequential follow-up questions based on the model’s prior responses, enabling the system to learn context-aware dialogue behavior. This design allows ExGra-Med to better support extended interactions that resemble clinical reasoning and decision-making processes.
>
> **Q3. Evaluations are limited to simple benchmarks (VQA-RAD, SLAKE, PathVQA) and primarily compared against LLaVA-Med. More challenging benchmarks such as OmmiMedVQA and recent SOTA medical multimodal LLMs are needed.**
>
> **A3**. We thank the reviewer for these valuable suggestions and fully agree that including more challenging benchmarks and broader comparisons strengthens the paper’s contributions.
> Fortunately, we have already included these evaluations but may not have highlighted them clearly enough in the main text:
>
> (i) The zero-shot results on **23 segmentation tasks** are actually sub-tasks **derived from the OmmiMedVQA benchmark** (Appendix Section G, Figure 4). Additionally, we conducted a **medical visual chatbot evaluation** (zero-shot) using **GPT-4 as the evaluator due to the complexity of the model responses** (Appendix Section F).
>
>
> (ii) We have also compared ExGra-Med with **state-of-the-art medical MLLMs** beyond LLaVA-Med, such as **RAD-FM** (pre-trained on 16M 2D/3D scans vs. 600k for LLaVA-Med), **MedDr** (40B parameters vs. our 7B model, trained on 2M instruction-tuning samples), and **GPT-4o API** for zero-shot evaluation on the medical visual chatbot task.
>
> Beyond medical MLLMs, we have **benchmarked the multi-graph alignment** module itself against advanced alignment algorithms across different domains such as **PLOT, SigLIP, VLAP (2-domain alignments)**, and **GeoCLAP, PAC-S, ImageBIND (3-modality alignments)**.
>
> In summary, we believe these diverse benchmarks (from standard to highly challenging) and comparisons with multiple SOTA baselines effectively validate ExGra-Med’s strengths. If the paper is accepted, we will revise the manuscript to emphasize these points more clearly to address the reviewer’s concerns.
>
> **Q4. Comparison with baselines trained using GPT-4-generated instruction-following data, not just used in multi-graph alignment**
>
> **A4**. We appreciate the reviewer’s suggestion and agree that clarifying this point is important. We would like to clarify that we have **already conducted analyses** to assess the effect of using GPT-4-generated extended instruction-following data beyond just multi-graph alignment.
> Specifically:
>
> (i) We trained **LLaVA-Med models using the GPT-4-extended instruction data**, and the results are presented in Table 3, denoted as LLaVA-Med (10%) ext. cap and LLaVA-Med (40%) ext. cap, representing training with 10% and 40% of the extended captions, respectively.
>
> (ii) For **all multi-modal pretraining algorithms in Tables 1 and 2**, including ExGra-Med and relevant baselines, the **same GPT-4-extended instruction-following data** were used to ensure a fair comparison. This setup helps isolate the effect of the multi-graph alignment itself, and results confirm the consistent improvements of ExGra-Med across all three VQA datasets.
>
>
> We will revise the manuscript to better highlight these evaluations and ensure this point is clearly communicated to readers.
>
>
> **Q5. Unclear whether the performance gains of ExGra-Med are robust across other foundation models beyond LLaVA-Med**
>
> **A5**: We thank the reviewer for raising this important point. To evaluate the robustness of ExGra-Med beyond the LLaVA-Med backbone, **we have integrated the multi-graph alignment framework to the Qwen-VL 7B model**, a strong vision-language foundation model.
>
> We compared the following three configurations:
> - Qwen-VL 7B baseline without additional pretraining on medical instruction-following data,
>
> - Qwen-VL 7B pre-trained with 10% GPT-4-enhanced medical instruction-following data, and
>
> - ExGra-Med uses the Qwen-VL 7B architecture, also pre-trained with the same 10% instruction-following data, but incorporating our multi-graph alignment strategy.
>
> All models were fine-tuned on the VQA-RAD and SLAKE datasets (closed-ended questions, evaluated with accuracy). Our results show that ExGra-Med consistently improves performance over both baselines on these benchmarks, with gains of 3–6%, demonstrating that our approach is model-agnostic and generalizes well across different foundation architectures.
>
> We will add these findings more explicitly in the revised manuscript to highlight the generalizability of our method.
>
> | Model                                           | VQA-RAD | SLAKE  |
> |------------------------------------------------|---------|--------|
> | LLaVA-Med (medical pre-trained 10%)            | 61.4%   | 80.3%  |
> | **ExGra-Med (LLaVA** architecture, pre-trained 10%)| 79.4%   | 85.10% |
> | Qwen-VL (general weights)                      | 53.85%  | 70.5%  |
> | Qwen-VL Med (medical pre-trained 10%)          | 52.73%  | 86.53% |
> | **ExGra-Med (Qwen-VL**, pre-trained 10%)           | 58.10%  | 89.75%    |

---

### Note · Authors · 2025-08-16

We sincerely thank all reviewers for their constructive feedback. We'd like to summarize our discussions as follows:

**A. Address Concerns of Reviewer Usn8**

- **Clarity & Reproducibility**: We will simplify the mathematical formulations and add visual explanations to improve accessibility. To ensure reproducibility and adoption, we will release pretrained checkpoints and the full training pipeline for multi-graph alignment.

- **Conversational Capability**: Although Figure 2 showed only a simplified case, ExGra-Med is trained on multi-turn conversational data (from LLaVA-Med), enabling context-aware dialogues that support clinical reasoning and decision-making.

- **Evaluation Scope**: Beyond VQA-RAD, SLAKE, and PathVQA, our evaluations already cover OmmiMedVQA subtasks, a medical chatbot benchmark (evaluated by GPT-4), and comparisons against SOTA medical MLLMs (e.g., RAD-FM, MedDr, GPT-4o). We also benchmark the multi-graph alignment module itself against advanced cross-domain alignment methods, confirming its broad applicability.

- **Fair Use of GPT-4** : All baselines and ExGra-Med used the same GPT-4-extended instruction-following data, ensuring fairness. Results across three VQA datasets confirm that the gains are due to multi-graph alignment, not just the instruction data.

- **Robustness Across Models**: We integrated ExGra-Med into Qwen-VL 7B, achieving 3–6% accuracy gains on VQA-RAD and SLAKE compared to both baseline and GPT-4-augmented Qwen-VL. This demonstrates that our approach is model-agnostic and generalizes well beyond LLaVA-Med.

**B. Confirmed addressed key concerns to other Reviewers Fpu5, xkmU, H5Kb.**

- **Backbone & Fairness**: Both ExGra-Med and LLaVA-Med use the same LLaA--7B (huggyllama/llama-7b), ensuring consistent and fair comparisons.

- **Robustness Across Models**: On Qwen-VL 7B/2.5, ExGra-Med achieved 3–6% accuracy gains on VQA-RAD and SLAKE, showing model-agnostic generalization.

- **Robustness to Caption Quality**: Using Qwen3-8B captions still yielded strong gains, confirming stability to caption source quality.

- **Triplet Alignment**: Ablations confirm the triplet setup (image, caption, extended caption) is superior to pairwise alignment.

- **Parameter-Efficient Fine-Tuning**: With LoRA, ExGra-Med still outperforms LLaVA-Med, though full fine-tuning remains stronger; larger-scale pretraining planned.

- **Additional Improvements**: We will add recent MLLM surveys to strengthen context in the final version.

---

### Decision · Program_Chairs · 2025-09-17

**Decision:**

Accept (poster)

**Comment:**

The paper received mixed ratings from the reviewers. Most of the comments are positive in terms of the technical design and the significance of model efficiency at the data level. One reviewer who gave a borderline rejection had concerns regarding the model design complexity and the additional experiments on larger benchmarks. The authors provided a comprehensive rebuttal on these issues. As these issues are not directly related to the core contribution, and all the other reviewers are satisfied with the paper's contribution, AC finally decided to recommend acceptance of the submission.